# Directed differentiation of mouse pluripotent stem cells into functional lung-specific mesenchyme

Andrea B. Alber [1,2], Hector A. Marquez[1,2], Liang Ma[1,2], George Kwong[1,2], Bibek R. Thapa[1,2], Carlos Villacorta-Martin[1], Jonathan Lindstrom-Vautrin[1], Pushpinder Bawa[1], Feiya Wang[1], Yongfeng Luo[3], Laertis Ikonomou [4,5], Wei Shi[6] & Darrell N. Kotton [1,2] ✉

While the generation of many lineages from pluripotent stem cells has resulted in basic discoveries and clinical trials, the derivation of tissue-specific mesenchyme via directed differentiation has markedly lagged. The derivation of lung-specific mesenchyme is particularly important since this tissue plays crucial roles in lung development and disease. Here we generate a mouse induced pluripotent stem cell (iPSC) line carrying a lung-specific mesenchymal reporter/lineage tracer. We identify the pathways (RA and Shh) necessary to specify lung mesenchyme and find that mouse iPSC-derived lung mesenchyme (iLM) expresses key molecular and functional features of primary developing lung mesenchyme. iLM recombined with engineered lung epithelial progenitors self-organizes into 3D organoids with juxtaposed layers of epithelium and mesenchyme. Co-culture increases yield of lung epithelial progenitors and impacts epithelial and mesenchymal differentiation programs, suggesting functional crosstalk. Our iPSC-derived population thus provides an inexhaustible source of cells for studying lung development, modeling diseases, and developing therapeutics.

The lung mesenchyme plays important roles in lung development and disease, yet knowledge is limited about the biology of lung mesenchymal progenitors, or how they initiate disease. In the mouse, lineage specification of the respiratory mesenchyme from splanchnic mesodermal precursors has been reported to occur around embryonic day (E) 9–9.5, a process coordinated through bi-directional endodermal-mesenchymal signaling interactions[1–4]. Mouse models have helped to identify several key regulators potentially involved in lung mesenchyme specification and differentiation, such as Retinoic Acid (RA), Wnt/β-catenin, Sonic hedgehog (SHH), and BMP4[1,3,5–14]. However, our understanding of early lung mesenchyme specification and differentiation in the mouse is still limited, and how the lung mesenchyme is specified during human development is unknown.

Complementing animal models, induced pluripotent stem cell (iPSC) or embryonic stem cell (ESC)-derived lineages have emerged as powerful tools to study both development and disease, since these in vitro models allow easy experimental manipulation and provide an unlimited source of material[15,16]. iPSCs/ESCs have the potential to generate any lineage of interest in vitro by directed differentiation, i.e., the in vitro recapitulation of the sequence of in vivo developmental

[1]Center for Regenerative Medicine of Boston University and Boston Medical Center, Boston, MA 02118, USA. [2]The Pulmonary Center and Department of Medicine, Boston University School of Medicine, Boston, MA 02118, USA. [3]Department of Surgery, Children's Hospital Los Angeles, Keck School of Medicine, University of Southern California, Los Angeles, CA 90027, USA. [4]Department of Oral Biology, School of Dental Medicine, University at Buffalo, Buffalo, NY 14260, USA. [5]Division of Pulmonary, Critical Care and Sleep Medicine, Department of Medicine, Jacobs School of Medicine and Biomedical Sciences, University at Buffalo, Buffalo, NY 14215, USA. [6]Division of Pulmonary, Critical Care, and Sleep Medicine, Department of Internal Medicine, University of Cincinnati College of Medicine, Cincinnati, OH 45267, USA. ✉e-mail: dkotton@bu.edu

milestones by stepwise addition of growth factors or small molecules to stimulate key developmental signaling pathways. Our lab and others have established protocols for the directed differentiation of both mouse and human iPSCs/ESCs into Nkx2-1+ lung epithelial progenitors, as well as toward more mature distal and proximal epithelial respiratory lineages[17–28]. In contrast, only two research groups have attempted to generate lung-specific mesenchyme by directed differentiation through a mesodermal progenitor[2,9,29]. While these studies have found an upregulation of lung mesenchymal markers by RT-qPCR in bulk cell extracts, they lacked specific reporters for cell tracking or purification, leaving unanswered questions about the efficiency of the differentiation protocol and the transcriptomic similarity of in vitro differentiated cells to primary lung mesenchyme.

A potential alternative approach to directed differentiation through a mesodermal progenitor state is the co-development of lung mesenchymal progenitors during in vitro differentiation of lung epithelial progenitor cells. Mouse lung epithelial differentiation protocols do not yield 100% differentiation efficiency, and thus might also contain non-epithelial lineages, such as mesodermal progenitors. Thus, co-development might occur when differentiating lung epithelial cells provide the signals necessary to specify lung mesenchyme from these mesodermal progenitor cells and vice versa. For example, we previously differentiated a lung epithelial-specific Nkx2-1[mCherry] reporter mouse ESC line into mCherry+ lung epithelium and observed emergence of early lung mesenchymal markers enriched in Nkx2-1[mCherry]- cells, implying co-development of both lineages[27]. Similarly, the emergence of human cells expressing general mesenchymal markers has been observed as a "by-product" of epithelial differentiation when using directed differentiation protocols without sorting for the epithelial progenitors of interest[17,30,31]. However, it remains unclear whether these mesenchymal cells can be classified as lung-specific mesenchyme, since no lung mesenchyme-specific reporters were available for use in these studies.

Importantly, several lines of evidence have suggested that embryonic mesenchyme is tissue-specific. Using tissue "recombinants", i.e., ex vivo cultures of embryonic epithelium mechanically combined with mesenchymal tissues, a study by Shannon has shown that recombining tracheal (i.e., proximal) epithelium with distal lung mesenchyme induced branching and expression of distal epithelial markers, while recombining lung (i.e., distal) epithelium with tracheal mesenchyme leads to the formation of cystic structures and expression of proximal epithelial markers[32]. This demonstrates that embryonic mesenchyme can determine region-specific epithelial identity and suggests different underlying gene expression profiles between region-specific embryonic mesenchymes. Supporting these early findings, Han et al. recently performed single cell RNA sequencing (scRNA-seq) of multiple organs of the early embryonic foregut and found specific transcriptional signatures for mesenchymes of each organ[2], highlighting the importance of generating tissue-specific mesenchyme as opposed to generic mesenchymal cells when using in vitro approaches to study development and disease.

While previous attempts to generate, track, and purify lung-specific mesenchyme have been limited by the lack of specific reporters or markers, this hurdle has been surmounted recently by the development of an in vivo lineage tracer/reporter system based on a lung mesenchyme-specific enhancer region in the *Tbx4* gene. While the *Tbx4* locus and transcript are more broadly expressed, this enhancer region appears to be selectively active in the developing lung mesenchyme and is evolutionarily conserved among multiple air-breathing species including mouse and human[33]. Zhang et al. and Kumar et al. have both successfully generated lung mesenchyme-specific lineage tracing and reporter mice using this *Tbx4* lung enhancer element[34,35], providing invaluable tools to study early lung mesenchyme development.

Here we engineer iPSCs derived from a *Tbx4* lung enhancer reporter/tracer mouse to develop a protocol for the directed differentiation of iPSCs into early lung mesenchyme. These iPSC-derived lung mesenchymal progenitors share key transcriptional features with primary mouse embryonic lung mesenchyme and are competent to differentiate into more mature mesenchymal lineages. We assess the functional capacity of these engineered lung mesenchymal progenitors by generating three-dimensional recombinant lung organoids composed of juxtaposed layers of separately derived engineered lung mesenchyme and engineered lung epithelium. We find that recombination appears to augment lung epithelial lineage specification and yield while impacting the molecular phenotypes of both lineages, suggesting functional epithelial-mesenchymal crosstalk in a multi-lineage lung organoid model system.

## Results

### Lung mesenchyme can be generated in vitro by directed differentiation

To develop a putative lung mesenchyme-specific in vitro fluorescent reporter/tracer system able to track and quantify the acquisition of early lung mesenchymal fate, we sought to generate an iPSC line from mice that we previously engineered to provide transgenic spatio-temporal lineage tracing of lung mesenchyme[35]. These triple transgenic mice carry an rtTA/LacZ encoding cassette under regulatory control of a *Tbx4* lung-specific enhancer, a Tet-responsive Cre recombinase, and a fluorescent mTmG cassette (Tbx4-rtTA; TetO-Cre; mTmG, hereafter "Tbx4-LER"), enabling inducible green fluorescence lineage tracing specifically of embryonic lung mesenchymal cells, without tagging of mesenchyme in other tissues in vivo[35] (Fig. 1a). To ensure that GFP tagging of primordial lung mesenchyme in these mice is specific and visible at the time of initial lung budding, a time point earlier than we have previously examined[35], we administered doxycycline (dox) to Tbx4-LER mice from embryonic day (E) E6.5 to E10 and observed GFP fluorescence at E10 in the developing primordial lung mesenchyme (Fig. 1a).

Next, we reprogrammed tail tip fibroblasts from a triple transgenic Tbx4-LER mouse, generating iPSCs that exhibited a normal karyotype and expressed pluripotency markers (Supplementary Fig. 1a–e). To test their mesenchymal competence, we first differentiated iPSCs into lateral plate mesoderm, the precursor tissue from which lung mesenchyme is thought to originate during early development. We used our previously published serum-free directed differentiation protocol[36] producing KDR+ putative mesodermal cells in 5 days in response to WNT3A, BMP4, and ACTIVIN A with similar efficiency to previously published studies[36,37] (Fig. 1b, c, Supplementary Fig. 1f, g). Consistent with our prior publication[36], sorted KDR+ day 5 cells were highly enriched in lateral plate mesoderm marker *Foxf1*, and expressed only low levels of paraxial and intermediate mesoderm markers *Tbx6* and *Pax2* (Fig. 1c). In the first 5 days of differentiation, no Tbx4-LER activation was observed in the presence of dox (Supplementary Fig. 1h), suggesting that day 5 iPSC-derived lateral plate mesoderm is not yet specified to a lung mesenchymal fate in these conditions.

In order to specify lung mesenchyme from lateral plate mesoderm cells, we tested the combinatorial effects of 4 growth factors and small molecules (Retinoic acid (RA), BMP4, WNT3A, and purmorphamine [PMA]) selected to stimulate developmental pathways with suggested roles in lung mesenchyme formation published in vivo[2]: RA, BMP, canonical Wnt, and Hedgehog (Hh) signaling, respectively. These 4 factors added beginning at day 5 together with dox resulted in 18.5 ± 5% of cells activating the Tbx4-LER[GFP] lineage trace by day 13 (Fig. 1d,e). To identify which of these factors is required for Tbx4-LER activation, we removed 1 or 2 factors at a time (Fig. 1e), finding that RA was absolutely required for Tbx4-LER[GFP] induction, whereas BMP4 was dispensable and removing both BMP4 and WNT3A resulted in the

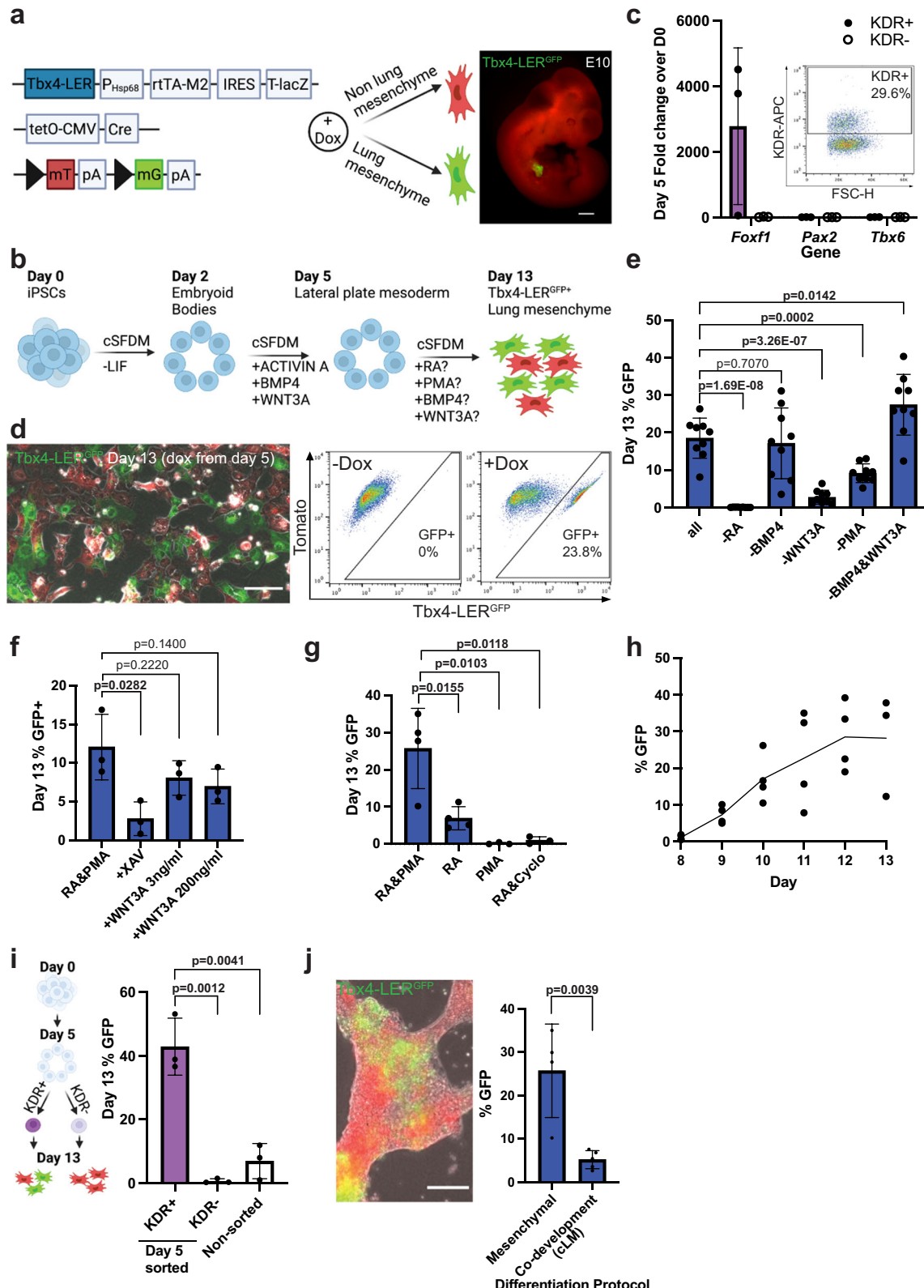

highest (27.4 ± 7.6%) Tbx4-LER^GFP efficiency, suggesting that stimulated RA together with Hh signaling represents a minimal combination required for efficient Tbx4-LER activation (Fig. 1e). Although supplemental WNT3A showed no dose-response effect (Fig. 1f) and indeed was not necessary for Tbx4-LER activation (Fig. 1e, condition -BMP4), inhibition of endogenous canonical Wnt signaling with XAV939 significantly reduced Tbx-LER^GFP induction (Fig. 1f), suggesting

that successful lung mesenchyme differentiation requires active endogenous Wnt signaling.

To further determine the differential effects of RA and Hh signaling on lung mesenchyme differentiation efficiency, we identified optimal concentrations of RA and PMA and found that the yield of Tbx4-LER^GFP+ cells peaks at 2 μM RA and 0.5 μg/ml PMA (Supplementary Fig. 1i,j). We next removed either RA or PMA from the

**Fig. 1 | In vitro differentiation of iPSCs toward the lung mesenchyme lineage through a mesodermal progenitor. a** Schematic of the *Tbx4* lung enhancer reporter/tracer (LER) line, and image of an E10 embryo (dox from E6.5 to E10). Scale bar = 0.5 mm. Created with BioRender.com. **b** Directed differentiation of iPSCs into lung mesenchyme through a mesodermal progenitor. cSFDM = complete serum-free differentiation medium, LIF = Leukemia inhibitory factor. Created with BioRender.com. **c** Expression relative to day 0 iPSCs of *Foxf1* (lateral plate mesoderm), *Pax2* (intermediate mesoderm) and *Tbx6* (paraxial mesoderm) in KDR± cells on day 5. *N* = 3. **d** Image and flow cytometry plot showing expression of GFP (green) in the Tbx4-LER iPSC line on day 13. Red = Tomato in non-recombined cells. Scale bar = 100 μm. **e** GFP percentage on day 13 using all 4 medium factors (RA, PMA, BMP4, WNT3A), or one or two factors removed at a time. Dox from day 5 on. *N* = 9. **f** Day 13 GFP percentage in RA&PMA medium, with either the XAV or recombinant mouse WNT3A. Dox from day 5 on. *N* = 3. **g** GFP percentage on day 13 in RA&PMA medium,

in RA or PMA only medium, or upon addition of Cyclopamine. Dox from day 5 on. *N* = 4 for "RA&PMA" and "RA", *N* = 3 for "PMA" and "RA&Cyclo". **h** GFP percentage over time from day 5 to 13 of differentiation. Dox from day 5 on. *N* = 4 for day 8-12, *N* = 3 for day 13. **i** GFP percentage on day 13 in RA&PMA medium after sorting for KDR± on day 5 of differentiation. *N* = 3. Created with BioRender.com.
**j** Representative image showing reporter activation on day 13 of lung epithelial differentiation and GFP percentage in the mesenchymal versus epithelial (i.e., co-development, cLM) differentiation protocol. Dox from day 5 on in the mesenchymal differentiation protocol, and from day 6 (anterior foregut endoderm stage) on in the epithelial (co-development) protocol. Scale bar = 100 μm. *N* = 4 for "Mesenchymal", *N* = 5 for "Co-development". All bars show mean ± sd. *p* values were determined by unpaired, two-tailed Student's *t* test. Significant (*p* < 0.05) *p* values are highlighted in bold font.

differentiation medium. Removing PMA significantly decreased the percentage of Tbx4-LER^GFP+ cells, but still generated an average of 6.9 ± 2.7% of Tbx4-LER^GFP+ cells (Fig. 1g). In contrast, removing RA led to a complete lack of Tbx4-LER^GFP+ cells, confirming that RA signaling is necessary to specify lung mesenchyme from lateral plate mesoderm (Fig. 1g). Finally, adding the Hh inhibitor Cyclopamine to RA-containing differentiation medium completely abolished Tbx4-LER^GFP activation (Fig. 1g), suggesting that combinatorial RA together with Hedgehog signaling promotes lung mesenchymal differentiation.

Having established a defined minimal medium (RA and PMA) for Tbx4-LER activation, we profiled the temporal dynamics of Tbx4-LER^GFP induction, adding dox from day 5 of differentiation on, and scoring Tbx4-LER^GFP efficiency daily until day 13. The percentage of Tbx4-LER^GFP+ cells increased from day 8 to day 12 of differentiation (Fig. 1h). Furthermore, sorting day 5 KDR+ cells enriched for GFP competence, yielding an average of ~40% of Tbx4-LER^GFP+ cells on day 13, whereas KDR- sorted cells were completely depleted of GFP competence, suggesting that only KDR+ lateral plate mesodermal cells serve as precursors for Tbx4-LER^GFP+ lung mesenchyme in our model system (Fig. 1i).

Previous studies have demonstrated that it is possible to "co-develop" both mesodermal and endodermal epithelial lineages together in the same differentiation medium when using directed lung epithelial differentiation protocols[17,27,30,31]. Hence, we aimed to investigate whether Tbx4-LER^GFP+ lung mesenchyme can also be derived by co-development during directed lung endodermal/epithelial differentiation as an alternative approach to directed differentiation via a mesodermal progenitor. We used our published lung epithelial differentiation protocol[27,38] to differentiate the Tbx4-LER iPSC line into the Nkx2-1+ lung epithelial progenitor stage (Supplementary Fig. 1k) and by day 13 observed an average of 5.2 ± 1.9% of Tbx4-LER^GFP+ cells (Fig. 1j), suggesting that co-development of lung mesenchyme ("cLM") and epithelium is indeed feasible. However, since lung mesenchymal differentiation by co-development was significantly less efficient than directed lung mesenchymal differentiation through a lateral plate mesoderm progenitor state ("mesenchymal"; Fig. 1j), we largely focused on directed lung mesenchymal differentiation through a mesodermal progenitor for further characterization and downstream applications.

### Induced lung mesenchyme expresses multiple lung mesenchymal markers

Next, we used scRNA-seq to compare the molecular phenotypes of Tbx4-LER^GFP+ cells and Tbx4-LER^GFP- cells generated by directed lung mesenchymal differentiation through a mesodermal progenitor state. For a control comparator we also profiled primary mouse embryonic lung mesenchyme from embryonic day (E) 12.5, (hereafter referred to as "primary") (Fig. 2a, b). By Louvain clustering (resolution = 0.1; Fig. 2b and Supplementary Fig. 2a) we found that iPSC-derived cells largely segregated into two distinct cell clusters, one almost exclusively

consisting of Tbx4-LER^GFP+ cells (cluster 1; Fig. 2b), the second mostly consisting of Tbx4-LER^GFP- cells with a minor subset composed of GFP+ cells (cluster 2, Fig. 2b). Primary lung mesenchymal cells clustered separately from iPSC-derived cells and formed two distinct clusters: a main mesenchymal cluster (cluster 3) and an additional minor cluster annotated as mesothelial cells based on canonical markers (cluster 4, Fig. 2b, Supplementary Fig. 2b,c, Supplementary Data 1). As expected, multiple primary developing lung mesenchymal sub-lineages (as described by others) could be identified within cluster 3 when using higher Louvain clustering resolutions (e.g., resolution = 0.5; Supplementary Fig. 2a).

Both the iPSC-derived cluster 1 and the major primary mesenchymal cluster 3 expressed lung mesenchymal markers, such as *Tbx4*, *Foxf1*, and Hh targets (*Gli1*, and *Hhip*), as well as *Pdgfra*. In contrast most of these markers were expressed at lower levels in iPSC-derived cluster 2 (Supplementary Fig. 2d). Analysis of the combined expression of a list of 13 lung mesenchymal markers (Supplementary Data 2) known from the literature including *Foxf1*, *Tbx4/5*, *Wnt2/2b* and Hh targets[2,5,7,12,14,39,40] (further referred to as LgM gene set), as well as a published set consisting of 61 genes by Han et al. (Supplementary Data 2) found to be enriched in E9.5 lung mesenchyme[2] further showed that cluster 1 iPSC-derived cells are more transcriptionally similar to primary E12.5 lung mesenchyme than iPSC-derived cluster 2 cells (Fig. 2c). In addition, we also analyzed the expression of both gene sets in co-developed lung mesenchyme (cLM) and found significantly lower expression levels of these gene signatures in cLM compared to iPSC-derived Tbx4-LER^GFP+ cells generated through directed mesenchymal differentiation (Supplementary Fig. 2a, e, f). Consistent with these results, iPSC-derived lung-mesenchymal cells clustered closer to primary cells than either cLM or iPSC-derived non-lung mesenchymal cells based on expression of the LgM gene set (hierarchical clustering; Fig. 2d).

Next, focusing on differences between iPSC-derived cluster 1 cells and primary cells, we found *Wnt2* to be more highly expressed in primary cells, along with canonical Wnt signaling target genes, such as *Lef1* and *Snai1/2* (Supplementary Fig. 2b,d), suggesting that activation of the Wnt signaling pathway is lower in cluster 1 cells in these conditions compared to primary embryonic lung mesenchyme. Importantly, we found that both iPSC-derived cluster 1 mesenchyme and primary embryonic lung mesenchyme expressed only low to undetectable levels of markers for embryonic non-lung foregut mesenchyme such as esophageal/gastric markers *Wnt4, Msc* and *Barx1*[2] (Fig. 2e), as well as low to undetectable levels of more mature mesenchymal lineage markers, such as *Acta2/Myh11* (smooth muscle), *Plin1/Pparg* (adipocytes/lipofibroblasts), and *Col2a1* (chondrocytes) (Supplementary Fig. 2g), suggesting that iPSC-derived cluster 1 cells at this time point express an early lung mesenchymal progenitor molecular phenotype without significant maturation.

To further investigate the differences between iPSC-derived Tbx4-LER^GFP+ and − cells we performed a pairwise comparison of iPSC-

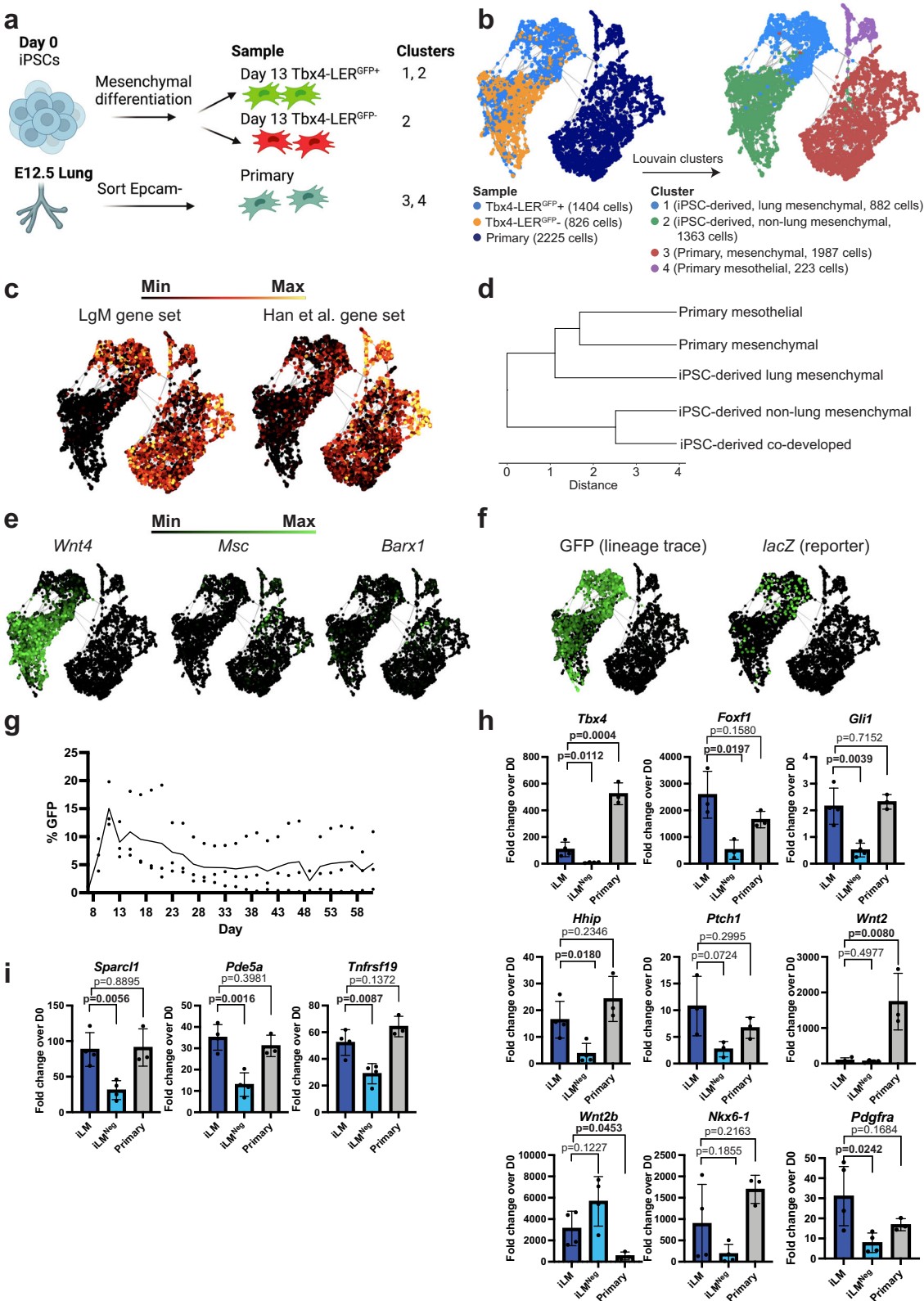

derived cluster 1 and cluster 2 cells (Supplementary Fig. 3a). While general mesenchymal markers were expressed in both clusters, we found that cluster 1 was defined by differentially expressed genes that included targets of the RA and Hh signaling pathways (*Cyp26b1*, *Ptch1*) as well as *Foxf1*, and several genes found to be enriched in early embryonic lung mesenchyme by Han et al., 2020, such as *Pde5a*, *Foxp2* and *Sparcl1*[2] (Supplementary Fig. 2b and 3a). In contrast, cluster 2

exhibited differentially increased expression of the embryonic eso-phageal marker *Wnt4*, as well as the mesothelial markers *Wt1* and *Upk3b* (Fig. 2e, Supplementary Fig. 2b,c, Supplementary Fig. 3a), sug-gesting that while cluster 2 cells are mesenchymal, they are less lung-like than cluster 1 cells. While *Tbx4* expression was enriched in Tbx4-LER[GFP]+ cells, it was also expressed in a subset of Tbx4-LER[GFP]- cells (Supplementary Fig. 3b), as expected for a reporter/tracer that only

**Fig. 2 | Transcriptomic characterization of iPSC-derived lung mesenchyme compared to primary embryonic lung mesenchyme. a** Schematic of the samples of interest included in scRNA-seq dataset. Created with BioRender.com. **b** SPRING plots showing the samples included in scRNA-seq analysis, and clusters identified by Louvain clustering (resolution 0.1). Tbx4-LER$^{GFP}$ ± cells were collected on day 13 of differentiation, dox from day 5 on. Primary = E12.5 embryonic lung mesenchyme. **c** Enrichment of a set of 13 embryonic lung mesenchyme markers known from the literature (LgM), and a set of 61 genes enriched in E9.5 lung mesenchyme published by Han et al., 2020. **d** Dendrogram showing unsupervised hierarchical clustering of primary populations (Primary mesenchyme and primary mesothelium), iPSC-derived lung mesenchymal, iPSC-derived non-lung mesenchymal and co-developed lung mesenchyme based on the LgM gene set. **e** Expression of embryonic esophageal/gastric markers *Wnt4*, *Msc*, and *Barx1* in iPSC-derived and primary cell clusters. **f** Expression of GFP and LacZ in iPSC-derived and primary cell clusters. **g** Dox induction time-course showing percentage of GFP+ cells over time,

with dox added 2 days before each day of quantification. *N* = 3. **h** RT-qPCR showing fold change expression relative to day 0 iPSCs of lung mesenchyme markers from the LgM gene set in induced lung mesenchyme (iLM) and iLM$^{Neg}$ cells compared to primary lung mesenchyme from E12.5 embryos. Tbx4-LER$^{GFP}$ ± cells were collected on differentiation day 13 and dox was added from day 5 on. *N* = 3 for "Primary" (all samples) and "iLM" and "iLM$^{Neg}$" for *Foxf1* and *Ptch1*, *N* = 4 for "iLM" and "iLM$^{Neg}$" for *Tbx4*, *Gli1*, *Hhip*, *Wnt2*, *Wnt2b*, *Nkx6-1*. **i** RT-qPCR showing fold change expression relative to day 0 iPSCs of lung mesenchyme markers from Han et al.'s gene set in induced lung mesenchyme (iLM) and iLM$^{Neg}$ cells compared to primary lung mesenchyme from E12.5 embryos. Tbx4-LER$^{GFP}$± cells were collected on differentiation day 13 and dox was added from day 5 on. *N* = 3 for "Primary", *N* = 4 for "iLM" and "iLM$^{Neg}$". All bars show mean ± sd. *P* values were determined by unpaired, two-tailed Student's *t* test. Significant ($p < 0.05$) *p* values are highlighted in bold font.

reflects activation of the lung mesenchyme-specific enhancer element within the Tbx4 locus, rather than expression of the locus itself.

Since a minor subset of Tbx4-LER$^{GFP}$+ lineage traced cells clustered together with Tbx4-LER$^{GFP}$- cells (cluster 2) and were not enriched in lung mesenchymal markers (Fig. 2b,c, Supplementary Fig. 3a), we considered the possibility that some GFP lineage-traced cells undergoing directed lung mesenchymal differentiation only transiently acquire a lung mesenchymal progenitor state, with subsequent loss of lung specific mesenchymal marker expression despite persistence of the GFP lineage trace. We anticipated that GFP+ cells that expressed an active developing lung mesenchymal progenitor program could be distinguished based on either short term dox induced GFP lineage tracing or based on expression of the lacZ reporter driven by the transgenic Tbx4 lung enhancer (Fig. 1a). Indeed, we found the subpopulation of GFP+ lineage traced cells that expressed high levels of lung mesenchymal markers were also enriched in transcripts encoding the lacZ reporter, whereas those GFP+ cells in cluster 2 did not express lacZ, suggesting they had lost their prior lung mesenchymal program as indicated by loss of Tbx4 lung-specific enhancer activity (Figs. 1a, 2f). We thus performed a time-course experiment where we differentiated cells using the lung mesenchymal differentiation protocol and quantified the percentage of Tbx4-LER$^{GFP}$+ cells every 2 days until day 61, while only adding dox 2 days prior to each analysis. We found that Tbx4-LER$^{GFP}$ activation peaks at day 11 of differentiation and subsequently decreases to lower levels (Fig. 2g), consistent with the hypothesis that the lung mesenchymal progenitor state acquired during directed lung mesenchymal differentiation is transient.

Having demonstrated that iPSC-derived cells produced by our protocol and expressing an active Tbx4 lung specific enhancer element are enriched in lung mesenchymal markers we thus annotated Tbx4-LER$^{GFP}$+ cells as putative induced lung mesenchyme (hereafter "iLM") and Tbx4-LER$^{GFP}$− cells as "iLM$^{Neg}$". We sought to validate the expression of a selected subset of these lung mesenchymal markers in iLM, iLM$^{Neg}$, and primary E12.5 lung mesenchymal cells by RT-qPCR. We selected markers genes from both the LgM gene set (Fig. 2h), as well as a subset of early embryonic lung mesenchymal markers from the Han et al, 2020 gene set (Fig. 2i, Supplementary Fig. 3c). We found significant enrichment in *Tbx4*, *Foxf1*, *Gli1*, *Hhip*, *Sparcl1*, *Pde5a*, and *Tnfrsf19* in iLM versus iLM$^{Neg}$ cells. Additional markers from both gene sets (*Nkx6-1*, *Ptch1*, *Stmn1*, *Osr1*, *Rspo1*, *Bcl11a*, *Hmga2*, *Chd3*) were highly expressed in iLM cells at levels no different from primary controls, and their expression was also detected in and not significantly different from iLM$^{Neg}$ cells.

Importantly, while many lung mesenchymal markers were expressed at similar levels in iLM and primary lung mesenchyme, we did detect significant differences in expression of a subset of markers. For example, *Tbx4 and Wnt2* were expressed at significantly lower levels in iLM cells compared to E12.5 lung mesenchyme, whereas the

expression of *Wnt2b* was significantly higher. Consistent with these results the total module scores of both Han et al and LgM gene sets were slightly lower in iLM than in primary E12.5 mesenchyme (Supplementary Fig. 2f).

We detected no significant differences in the expression of the general mesenchymal markers *Pdgfra* and *Col1a1*, or the smooth muscle marker *Acta2* between iLM cells and our primary control (Fig. 2h, Supplementary Fig. 3d). In conclusion, many of the assessed mesenchymal genes were expressed at similar, though not identical, levels to primary control cells, further confirming observations from our scRNA-seq dataset, and indicating that iPSC-derived engineered lung mesenchyme purified based on Tbx4-LER activation exhibits key transcriptomic features reminiscent of primary embryonic lung mesenchyme.

Since our transcriptomic comparisons were based on analyses from a single time point, we next sought to investigate how temporal changes in mesenchymal transcriptomic programs in our engineered cells compare to embryonic time-points in vivo for developing lung mesenchyme. Thus, we collected Tbx4-LER$^{GFP}$+ cells every day from day 8 to 13 of lung mesenchymal differentiation (adding dox only 2 days before each collection), as well as primary mouse embryonic lung mesenchymal cells from days E11.5 (i.e., the earliest time-point where it is technically feasible to collect a sufficient number of cells for reliable RT-qPCR analysis), E12.5, E13.5 and E18.5. We quantified the expression of several embryonic lung mesenchymal markers, both from the LgM gene set and Han et al.'s gene set, as well as several general/more mature mesenchymal markers (Supplementary Fig. 3e). We found that the expression of most measured lung mesenchymal markers in Tbx4-LER$^{GFP}$+ cells peaked around day 10–12 of differentiation. Expression levels of 12 out of 15 markers were similar to E11.5–12.5 primary lung mesenchyme, but we observed important differences in *Wnt2* and *Tbx4*, which were lower in the Tbx4-LER$^{GFP}$+ cells at all time points assayed (Supplementary Fig. 3e), consistent with our scRNA-seq dataset (Supplementary Fig. 2b–d). We thus selected day 11 Tbx4-LER$^{GFP}$+ cells and E12.5 primary mesenchymal cells for subsequent co-culture experiments (see Figs. 3–6).

### Induced lung mesenchymal progenitors have developmental competence

We next sought to investigate whether iLM cells are competent to differentiate into more mature mesenchymal lineages (Supplementary Fig. 4a), including those that are not lung specific, such as smooth muscle, fat, or endothelium. We sorted day 13 Tbx4-LER$^{GFP}$+ versus Tbx4-LER$^{GFP}$- cells and re-plated them in a smooth muscle differentiation medium that we previously published[36]. After 4 days, both populations of cells elongated and significantly upregulated expression of smooth muscle markers *Acta2* and *Tagln*, while downregulating expression of *Tbx4* (Supplementary Fig. 4b–d), indicating mesenchymal maturation competence of both GFP+ and GFP− cells.

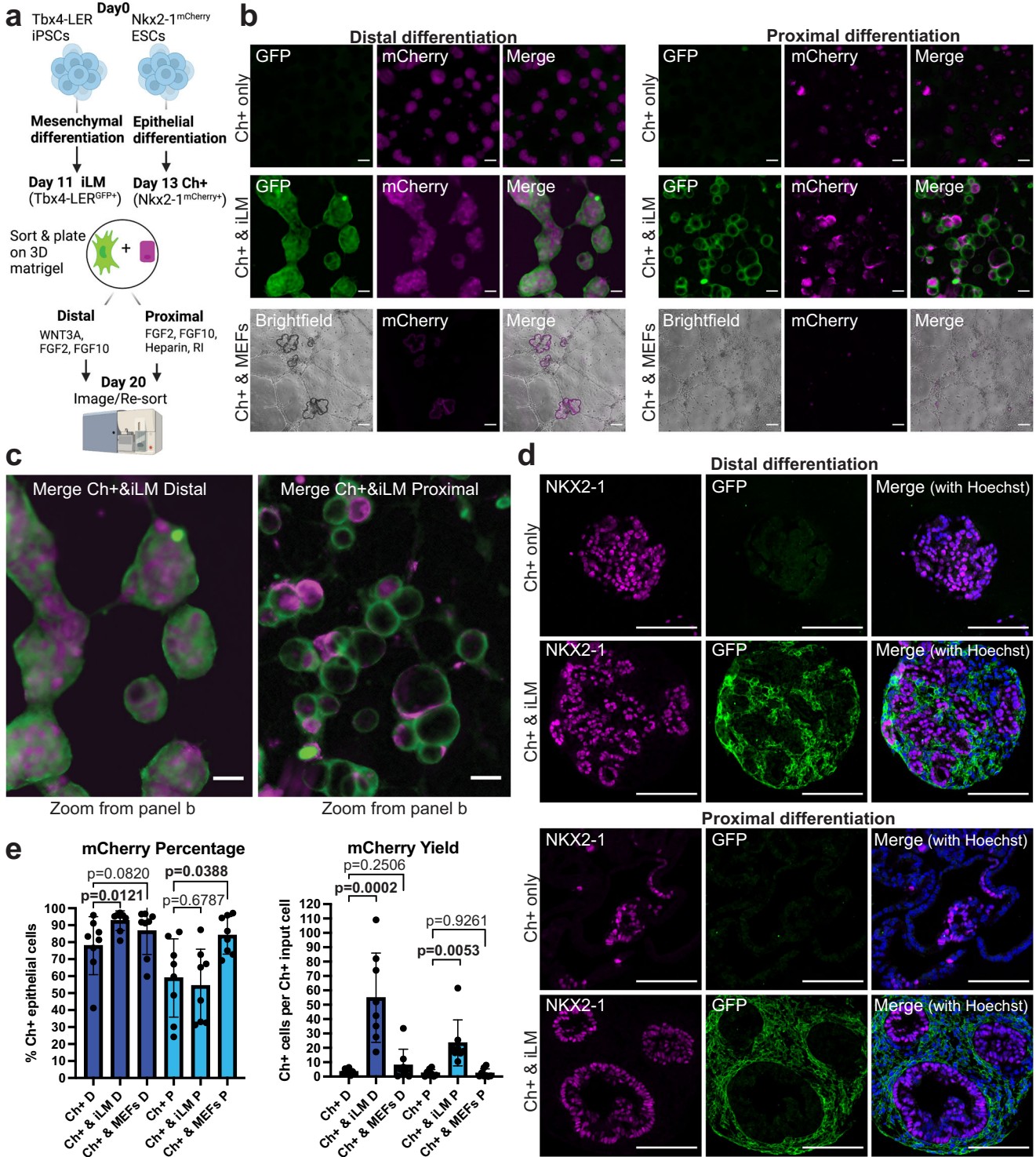

**Fig. 3 | Co-culture of induced lung mesenchyme with Nkx2-1mCherry+ lung epithelial progenitors leads to the formation of three-dimensional organoids and increases yield of Nkx2-1mCherry+ cells in distal and proximal epithelial differentiation conditions. a** Schematic of lung epithelial-mesenchymal co-culture experiments. Nkx2-1mCherry ESCs and Tbx4-LER iPSCs were differentiated separately using the lung epithelial or lung mesenchymal differentiation protocol. Day 11 Tbx4-LERGFP + (=iLM) and Day13 Nkx2-1mCherry + (Ch + ) cells were then mixed at a 20:1 ratio and cultured for 7 days in either distal or proximal differentiation medium. Created with BioRender.com. **b** Representative images of day 20 co-cultures. Day 13 Nkx2.1mCherry + (Ch + ) cells were cultured in distal or proximal medium, either alone,

with iLM cells or with a non-lung mesenchyme control (Mouse embryonic fibroblasts, MEFs). Scale bars = 200 μm. **c** Zoomed in versions of distal and proximal co-culture images shown in panel b. Scale bars = 200 μm. **d** Immunofluorescence images showing expression of NKX2-1 and GFP in Nkx2-1mCherry+ cells cultured in distal or proximal conditions alone or co-cultured with iLM for 1 week. Scale bars = 100 μm. **e** Percentage of Nkx2-1mCherry+ cells and yield (i.e., number of Nkx2-1mCherry+ cells per day 13 Nkx2-1mCherry+ input cell) for Nkx2-1Cherry+ cells cultured alone, with iLM or with MEFs in distal or proximal medium. *N* = 8. Bars show mean ± sd. *P* values were determined by paired, two-tailed Student's *t* test. Significant (*p* < 0.05) *p* values are highlighted in bold font.

Consistent with this general mesenchymal competence, we found an increased deposition of lipid droplets when culturing iLM cells in adipogenic medium, as compared to parallel control cultures of iLM cells continued in mesenchymal medium (Supplementary Fig. 4e). In contrast, iLM cells by day 7 had lost vascular endothelial differentiation competence. When day 5 iPSC-derived lateral plate mesodermal cells were re-plated in VEGF-containing endothelial medium, the endothelial marker CDH5 was later expressed on the surface of 20% of the cellular outgrowth, whereas differentiating unsorted Tbx4-LER cells transferred to the same endothelial medium on day 7 gave rise to few (<5%) CDH5+ endothelial cells when analyzed after 2–8 more days in culture, and none of these emerging CDH5+ cells were Tbx4-LER[GFP] + (Supplementary Fig. 4f). This suggests rapid loss of endothelial competence during directed lung mesenchymal differentiation after the lateral plate mesoderm state. These findings are in agreement with Zhang et al., who showed that developing Tbx4-LER[GFP]+ cells rapidly lose endothelial competence in vivo by E11.5, while their competence to differentiate into other mesenchymal lineages is maintained until later embryonic time-points[35].

### Induced lung mesenchyme co-cultured with engineered lung epithelium forms 3D organoids

One of the key functions of developing lung mesenchyme in vivo is to closely interact with and signal to the developing lung epithelium. Thus, in order to investigate the functional competence of iLM, we established a lung epithelial-mesenchymal co-culture system (Fig. 3a) where we separately differentiated our Tbx4-LER[GFP] iPSC line into iLM cells, and a Nkx2-1[mCherry] reporter ESC line[41] into an Nkx2-1+ lung epithelial progenitor state using our published protocol[27,38]. As noted above, in order to enrich for cells that have an activated Tbx4 lung-specific enhancer element at the time of collection (see Fig. 2f, g), we used only short term dox exposure, restricting the addition of dox to differentiation day 9–11, and collected Tbx4-LER[GFP]+ cells on day 11 for all co-culture experiments. We purified lung epithelial and iLM derivatives by flow cytometry and plated them on top of a bed of 3D Matrigel in our previously published distal (alveolar) or proximal (airway) lung epithelial differentiation media[27,38] (Fig. 3a).

Co-cultured cells self-organized into three-dimensional organoids (recombinants) with juxtaposed layers of Tbx4-LER[GFP]+ lung mesenchyme and Nkx2-1[mCherry]+ lung epithelium (Fig. 3b–d). Distal and proximal cultures differed in morphology, with distal cultures showing a more lobular and proximal cultures showing a more cystic structure (Fig. 3b, c). In contrast, Nkx2-1[mCherry]+ lung epithelium co-cultured with mouse embryonic fibroblasts (MEFs, i.e., a non-lung mesenchymal cell type control) did not associate closely and showed poor outgrowth of Nkx2-1[mCherry]+ cells, suggesting that co-culture with non-lung mesenchymal cells does not lead to 3D self-organization or detectable physical juxtaposition of the two lineages (Fig. 3b). Re-sorting after co-culture to analyze the frequency and yield of each cell lineage revealed that co-culture with iLM significantly increased both the percentage and yield of Nkx2-1[mCherry]+ epithelial cells after 1 week in distal differentiation medium, as compared to Nkx2-1[mCherry]+ only culture controls (Fig. 3e, Supplementary Fig. 5a). Furthermore, co-culture with iLM significantly increased the yield of Nkx2-1[mCherry]+ cells in proximal culture conditions compared to Nkx2-1[mCherry]+ cells cultured alone, while co-culture with MEFs did not (Fig. 3e).

### Co-culture with induced lung mesenchyme affects epithelial differentiation programs

We next sought to determine whether co-culture with iLM might provide signals that impact the molecular phenotype of differentiating ESC-derived lung epithelial cells. Given that iLM resembles an early lung mesenchymal progenitor state, we speculated that co-culture with iLM might maintain Nkx2-1[mCherry]+ cells in a less mature state compared to Nkx2-1[mCherry]+ epithelial-only cultures. Indeed, RT-qPCR

analysis of re-sorted cells indicated that Nkx2-1[mCherry]+ cells co-cultured with iLM in distal medium expressed higher levels of early lung epithelial markers such as *Sox9* and *Etv5*, but lower levels of more mature distal epithelial lineage markers such as *Sftpc*, *Ager* and *Hopx*, compared to Nkx2-1[mCherry]+ cells cultured alone (Fig. 4a, Supplementary Fig. 5b).

In proximal differentiation conditions, expression of airway markers known to arise early in development, such as *Sox2* and *Trp63* was increased in Nkx2-1[mCherry]+ cells co-cultured with iLM compared to Nkx2-1[mCherry]+ cells cultured alone. In contrast, markers that arise later in development, such as the club cell marker *Scgb1a1*, were decreased in Nkx2-1[mCherry]+ cells co-cultured with iLM compared to Nkx2-1[mCherry]+ cells cultured alone (Fig. 4a, Supplementary Fig. 5b). Nkx2-1[mCherry]+ cells in all conditions tested were enriched for *Nkx2-1*, but expressed little to no thyroid marker *Pax8* or forebrain marker *Otx1* (Supplementary Fig. 5c), suggesting maintenance of lung-specific identity. Immunofluorescence further confirmed expression at the protein level of *Sox9*, *Sftpc*, *Sox2*, and *Trp63* and indicated that in recombinant cultures GFP lineage-traced iLM descendants surrounded and abutted the more centrally located lung epithelia expressing these markers (Fig. 4b).

To validate our recombinant model using primary developing lung mesenchyme, we repeated these experiments combining ESC-derived purified lung epithelial progenitors (Nkx2-1[mCherry] sorted) with either E12.5 primary lung mesenchyme (pLM) or iLM (Fig. 5a–c, Supplementary Fig. 5d) in distal medium. After 1 week of co-culture as recombinants, we observed closely juxtaposed epithelial-mesenchymal cell layers with similar yields of Nkx2-1[mCherry]+ cells, which expressed similar levels of *Sox9*, *Etv5*, and *Sftpc* regardless of whether the cells were co-cultured with engineered iLM or pLM, but significantly higher *Ager* expression in the Nkx2-1[mCherry]+ cells co-cultured with pLM. Taken together, our results suggest that co-culture with induced lung mesenchyme not only increases the yield of Nkx2-1[mCherry]+ cells after culture in distal and proximal differentiation medium, but also affects the epithelial differentiation programs of their descendants.

We next sought to investigate whether our iLM is able to compensate for removal of growth factors in our distal and proximal media. We found no outgrowth in our complete serum-free differentiation medium base (cSFDM; with no growth factors added) when culturing Nkx2-1[mCherry]+ cells alone, with iLM, or with pLM. These findings are in agreement with existing literature, since to our knowledge there are no previous reports of lung epithelial-mesenchymal co-cultures in serum-free, growth factor-free media. We thus performed a factor withdrawal experiment, where we removed one growth factor at a time from our distal or proximal media and quantified the yield of Nkx2-1[mCherry]+ cells after 1 week of culture alone or with iLM (Supplementary Fig. 5e). We found that co-culture with iLM can partially compensate for the removal of WNT3A and FGF2 from the distal and the removal of FGF10 and FGF2 from the proximal medium.

### Co-culture induces lung epithelial fate in non-lung epithelial progenitors

Having found that co-culture with iLM augments the percentage and yield of Nkx2-1[mCherry]+ epithelial progenitors in distal differentiation conditions, we sought to determine whether iLM can also actively induce lung epithelial fate in ESC-derived endodermal cultures. We thus sorted day 13 Epcam + /Nkx2-1[mCherry]- cells (i.e., epithelial cells that had not acquired a lung-specific fate by day 13 of differentiation) and co-cultured them with iLM in distal differentiation medium (Fig. 5d). The resulting recombinants self-assembled into 3D structures with similar morphology to recombinants of iLM with Nkx2-1[mCherry]+ cells (Fig. 5e). After 7 days, 82.0 ± 4.8% of the previously Nkx2-1[mCherry]- cells expressed mCherry (Fig. 5f). In contrast, only 27.3 ± 14.6% of Nkx2-1[mCherry]- cells cultured in distal differentiation medium alone turned

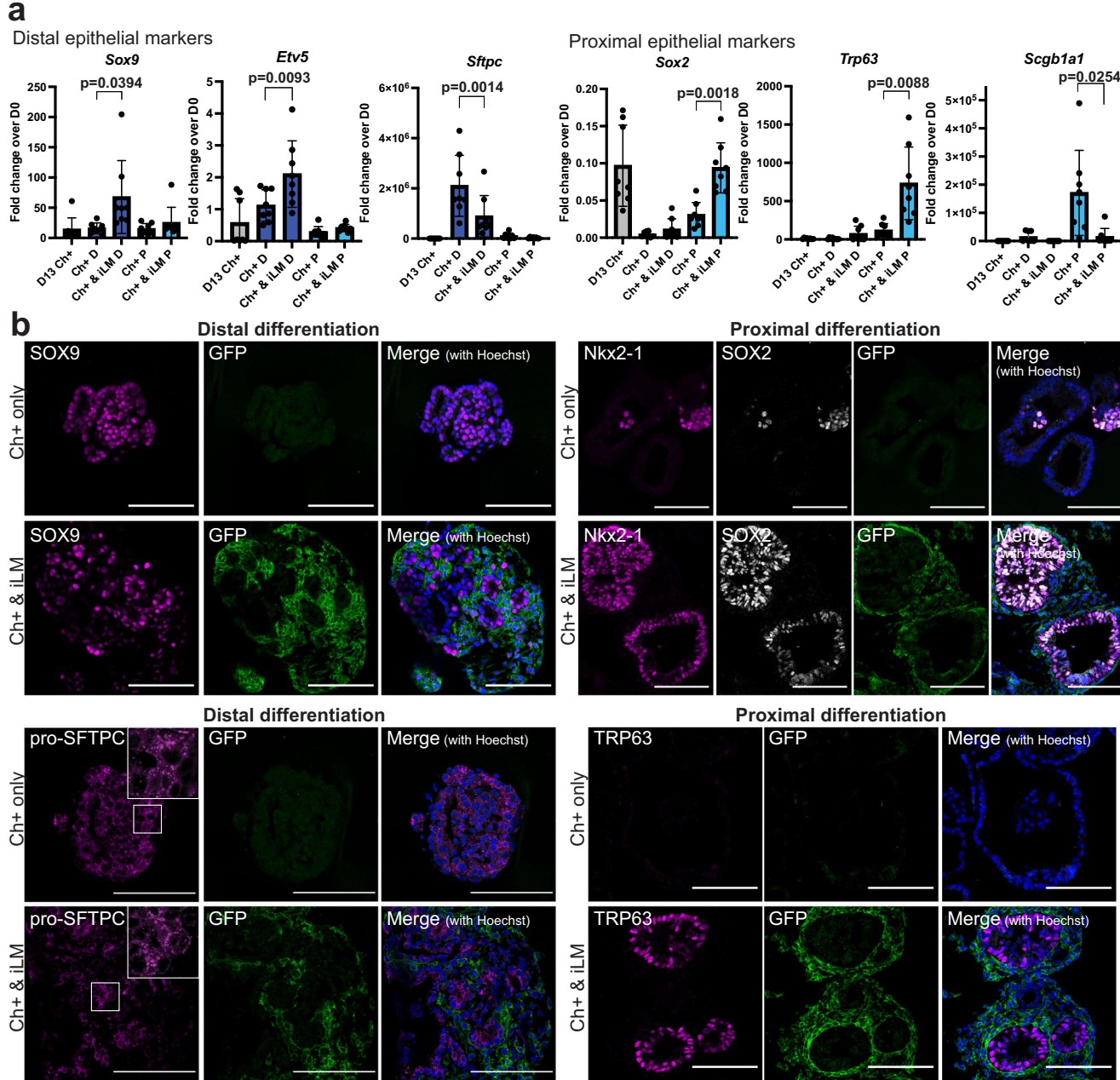

**Fig. 4 | Co-culture with induced lung mesenchyme impacts the differentiation program of Nkx2-1^mCherry+ lung epithelial progenitors. a** RT-qPCR data showing fold change expression relative to day 0 iPSCs of distal and proximal lung/airway markers in Nkx2-1^mCherry+ cells before (gray) and after co-culture in distal (dark blue) and proximal (light blue) medium. *N* = 8. Bars show mean ± sd. *P* values were determined by paired, two-tailed Student's *t* test. Significant (*p* < 0.05) *p* values are highlighted in bold font. **b** Immunofluorescence images showing expression of distal and proximal epithelial markers of interest in Nkx2-1^mCherry+ cells cultured in distal or proximal conditions alone or co-cultured with iLM for 1 week. Scale bars = 100 µm.

mCherry positive, and Nkx2-1^mCherry- cells co-cultured with MEF controls did not detectably expand in culture (Fig. 5f). Re-sorting of post co-culture Nkx2-1^mCherry+ cells and RT-qPCR analysis revealed no differences in expression of lung epithelial lineage markers in the Nkx2-1^mCherry+ outgrowth from any sample, regardless of whether the cells were Nkx2-1^mCherry+ or Nkx2-1^mCherry- prior to co-culturing with iLM (Supplementary Fig. 5f).

To further investigate the functional capacity of iLM cells, we tested their ability to support the outgrowth of primary embryonic or adult lung epithelial cells. First, we co-cultured iLM with purified primary E12.5 lung epithelial cells (pE, sorted based on our published Nkx2-1^GFP reporter[42]) in distal differentiation medium (Fig. 5g). Similar to our co-cultures with engineered lung epithelial progenitors, we found that co-cultures with iLM: (1) formed recombinants with closely juxtaposed layers of primary lung epithelium and mesenchyme (Figs. 5h), (2) significantly augmented yields of primary Nkx2-1^GFP+ epithelial cells after co-culture, compared to epithelial culture alone (Figs. 5i), and (3) resulted in augmented epithelial expression of *Etv5* but restrained expression of *Sftpc* or *Ager* (Fig. 5j). Second, we investigated whether our iLM can support outgrowth of adult primary mouse alveolar epithelial type 2 (AT2) cells (Fig. 5k). AT2 cells in prior reports grow well in co-culture with primary mouse lung mesenchymal cells[43], but do not survive in culture in these conditions in the absence of mesenchyme. We found that, similar to our co-cultures with engineered epithelial progenitors, AT2 cells purified from an adult Sftpc^tdTomato lineage

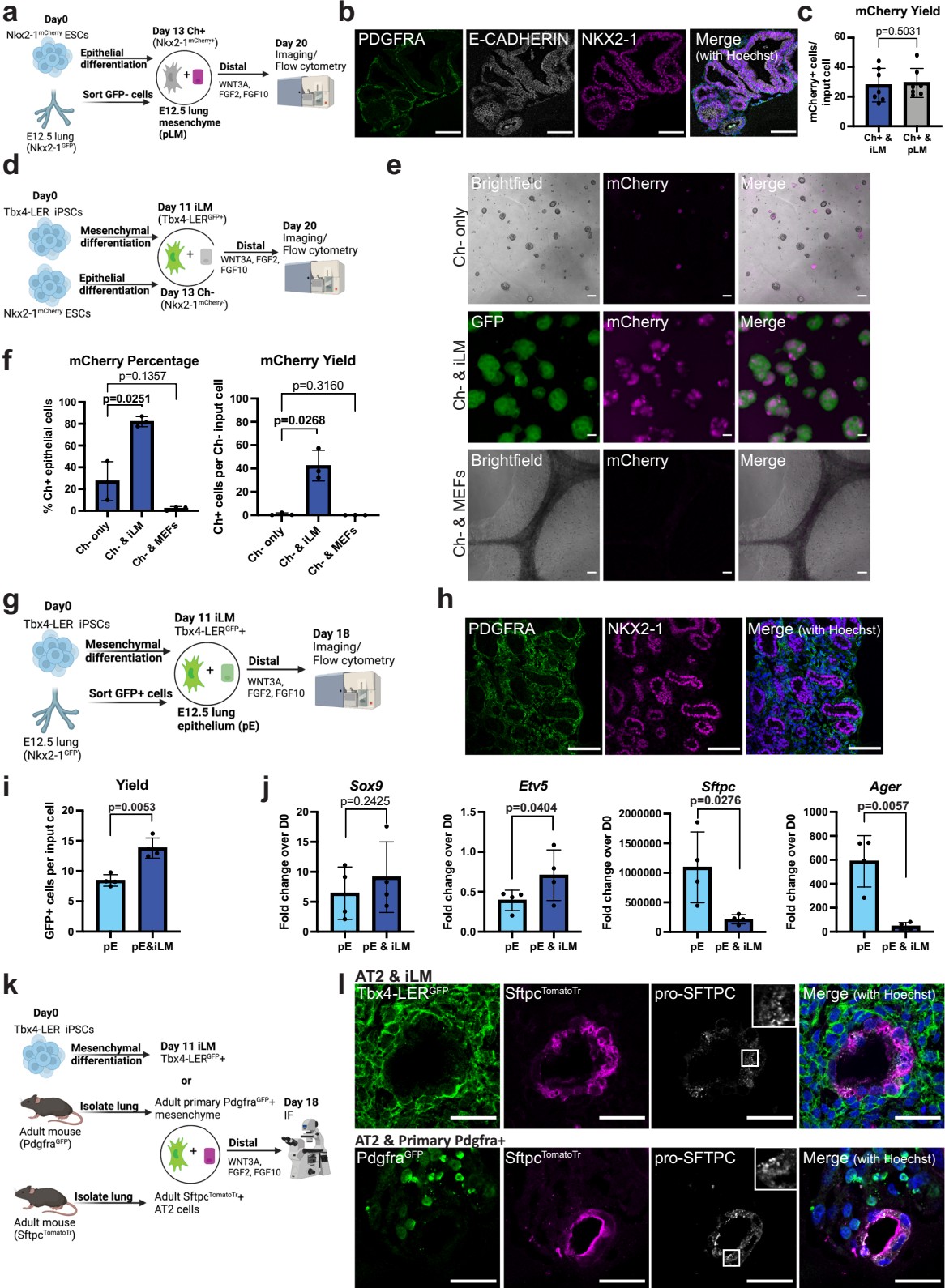

tracer mouse[44] and co-cultured with either iLM or with purified primary Pdgfra[GFP+] lung fibroblasts formed 3D organoids with closely juxtaposed layers of mesenchyme and proSFTPC+ epithelium (Fig. 5l). Thus, our data suggests that iLM can support outgrowth of both engineered and primary lung epithelial cells, including AT2 cells.

**Epithelial-mesenchymal crosstalk in co-cultures affects mesenchymal differentiation**

Finally, we sought to investigate how co-culture with Nkx2-1[mCherry+] cells affects the differentiation of iLM cells. Epithelial-mesenchymal interactions in lung development are thought to impact mesenchymal differentiation programs through bi-directional cross talk. However,

**Fig. 5 | Co-culture with in vitro differentiated lung mesenchyme can induce lung epithelial fate in non-lung epithelial progenitors and can maintain primary epithelial cells. a** Co-cultures of Nkx2-1mCherry+ cells with primary E12.5 lung mesenchyme (pLM). Created with BioRender.com. **b** Immunofluorescence showing expression of PDGFRA, E-CADHERIN and NKX2-1 in co-cultures of Nkx2-1mCherry+ epithelial cells and pLM in distal medium. Scale bars = 100 μm. **c** Yield of Nkx2-1Cherry+ cells after co-culture with iLM or pLM in distal medium N = 7. **d** Schematic of co-cultures of iLM with Nkx2-1mCherry- day 13 cells. Created with BioRender.com. **e** Representative images of co-cultures of iLM with Nkx2-1mCherry- cells. Cells were sorted for EPCAM +/mCherry− on day 13 of epithelial differentiation and cultured either alone, with iLM, or with non-lung mesenchyme (MEFs) in distal medium for 7 days. Scale bars = 200 μm. **f** Percentage and yield of Nkx2-1mCherry+ cells for Nkx2-1mCherry- cells cultured alone, with iLM or with MEFs in distal medium for 7 days. N = 3.

**g** Schematic of co-cultures of primary Nkx2-1GFP + E12.5 lung epithelial cells (pE) with iLM. Created with BioRender.com. **h** Immunofluorescence images showing expression of PDGFRA and NKX2-1 in co-cultures of pE and iLM in distal medium. Scale bars = 100 μm. **i** Yield of Nkx2-1GFP+ cells after 1 week of distal culture alone or with iLM. N = 4. **j** RT-qPCR data showing fold change expression over day 0 iPSCs of *Sox9*, *Etv5*, *Sftpc*, and *Ager* in pE cultured alone or with iLM in distal medium. N = 4. **k** Schematic of co-cultures of iLM with SFTPCTomatoTr lineage traced adult primary alveolar epithelial type 2 (AT2) cells. Created with BioRender.com. **l** Immunofluorescence images showing expression of GFP, Tomato and pro-SFTPC in primary SFTPCTomatoTr lineage traced AT2 cells cultured either with iLM or with primary PdgfraGFP+ lung mesenchyme. Scale bars = 30 μm. Bars show mean ± sd. P values were determined by paired, two-tailed (**c** and **f**) or one-tailed (**i** and **j**) Student's t test. Significant (p < 0.05) p values are highlighted in bold font.

few model systems are available to test this hypothesis in developing mammals. Supporting a critical role for the developing lung epithelium in sustaining mesenchymal cells in our model system, we observed that iLM cultured in distal differentiation medium alone showed poor survival and outgrowth. We thus first quantified the expression of lung mesenchymal markers in iLM cells before and after co-culture in order to test whether co-culture with differentiating lung epithelial progenitors alters mesenchymal programs (Fig. 6a–c). For non-lung mesenchymal controls, we quantified gene expression in MEFs after co-culture with ESC-derived Nkx2-1mCherry + (Ch + ) lung epithelium.

Co-culturing in distal medium with Nkx2-1mCherry+ lung epithelium resulted in upregulation of *Wnt2*, *Tbx4*, *Pdgfra*, and *Acta2* uniquely in iLM, but not in MEFs, while general mesenchymal marker *Col1a1* increased in all conditions in both iLM and MEFs (Fig. 6a). In contrast, lung mesenchymal progenitor marker *Foxf1* decreased. *Wnt2* is an important and relatively specific marker of both embryonic and adult lung mesenchymal lineages in vivo[1], and these results suggest that juxtaposed developing lung epithelium in distal conditions might provide important factors to maintain and augment or mature the lung-specific identity of co-cultured iLM.

Given the upregulation of *Acta2* transcript and expression of smooth muscle actin protein (SMA, also known as ACTA2; Fig. 6a, b) in iLM descendants in both distal and proximal recombinant co-cultures, we speculated that further maturation or patterning of lung mesenchyme may be occurring in our co-cultures. To explore this possibility, we first defined the in vivo global transcriptomes and cellular heterogeneity of all *Acta2*+ cells in the adult mouse lung at single cell resolution, using scRNA-seq to profile all GFP+ lung cells sorted from a transgenic mouse carrying bifluorescent Acta2hrGFP and Cspg4-Cre-LSL-dsRed reporters driven by promoter fragments for each gene[45] (Supplementary Fig. 6a). Dimensionality reduction visualization (SPRING) as well as Louvain clustering revealed that most *Acta2*+ lung cells segregate into 3 main clusters, one of which expressed *Notch3*, as well as the dsRed lineage trace, identifying this cluster as vascular smooth muscle cells[45] (Supplementary Fig. 6b–d). We further compared our dataset to a previously published scRNA-seq dataset of adult mouse lung mesenchymal cell populations[46] and identified our 3 main clusters as Mesenchymal alveolar niche cells (MANCs), airway smooth muscle (ASM) and vascular smooth muscle cells (VSM), based on an overlay of the top 50 genes (Supplementary Data 3) found to be enriched in these 3 cell types by Zepp et al.[46] (Supplementary Fig. 6c).

To determine whether these three cell types might be detectable in iLM recombinant cultures, we performed RT-qPCR of markers for each cell type in sorted iLM before and after recombinant cultures. We selected genes found among the top 30 enriched genes from our 3 in vivo cell clusters (Fig. 6d, Supplementary Fig. 7a, Supplementary Data 4). We also selected VSM marker *Timp4*, since it was found to be enriched in VSM cells in previous datasets[46,47]. Of note, most of the enriched genes selected from our mouse lung profiles were also found

among the top enriched genes in Zepp et al's datasets, suggesting that these are suitable markers for each of the 3 corresponding cell types. By RT-qPCR we found that the majority of MANC and ASM markers were upregulated in iLM after recombinant culture with epithelium, both in distal and proximal media compared to pre-co-culture iLM (Fig. 6d, Supplementary Fig. 7a). In contrast, expression of VSM markers was either significantly decreased or unchanged (Fig. 6d, Supplementary Fig. 7a), suggesting that co-cultured iLM might differentiate toward airway mesenchymal and MANC-like subpopulations, but not toward vascular smooth muscle cells, consistent with the presence of alveolar/airway-like epithelial cells, but lack of vascular endothelial cells in these co-cultures.

Finally, to obtain a more global view of gene expression in co-cultured iLM cells we performed scRNA-seq of iLM before and after co-culture with ESC-derived lung epithelial Nkx2-1mCherry+ cells in either distal or proximal media (Fig. 6e, Supplementary Fig. 7b). Consistent with the responses to co-culture profiled by RT-qPCR in Fig. 6a, we again observed increased expression of *Wnt2* after distal co-culture and augmented *Acta2* and *Col1a1* with decreased *Foxf1* and decreased expression of the early embryonic lung mesenchymal marker gene set, LgM (Fig. 6e, Supplementary Fig. 7b).

To investigate potential distal versus proximal patterning in our co-cultured lung mesenchyme, we employed proximal and distal adult lung mesenchymal marker datasets previously established by Wang et al., 2018[48] (Supplementary Data 5). As expected, we found that distal markers by Wang et al. tend to be expressed at higher levels in iLM after co-culture in distal medium, whereas proximal markers tend to be more enriched in iLM after proximal co-culture, (Fig. 6f, Supplementary Fig. 7c). Furthermore, co-culture resulted in upregulation in iLM of MANC and ASM gene sets from Zepp et al., 2021 (Supplementary Data 3; Fig. 6g). Integration of our iLM datasets with previous time series scRNA-seq profiles of developing lung mesenchyme by Zepp et al.[46] further demonstrated these changes in iLM after co-culture suggesting some degree of iLM maturation (Supplementary Fig. 8), although interpretations of these integrated datasets are potentially limited by technical batch effects. We also observed upregulation in iLM after co-culture of sets of the top 50 enriched genes of several adult mouse lung mesenchymal lineages defined by Tsukui et al, 2020[49] (peribronchial, alveolar, and adventitial fibroblasts; Supplementary Data 6; Fig. 6h). Consistent with RT-qPCR results (Fig. 6d), we did not observe significant upregulation of vascular mesenchymal gene sets, such as those of pericytes or vascular smooth muscle (Fig. 6g, Supplementary Fig. 8c).

## Discussion

Our results demonstrate the directed differentiation of mouse iPSCs into functional lung-specific mesenchymal cells via KDR+ mesoderm in response to combinatorial RA and Hh signaling. Utilizing a Tbx4 enhancer element that is uniquely active in the developing lung mesenchyme of air-breathing organisms we generated a murine lineage reporter/tracer iPSC line, and used this line to optimize defined,

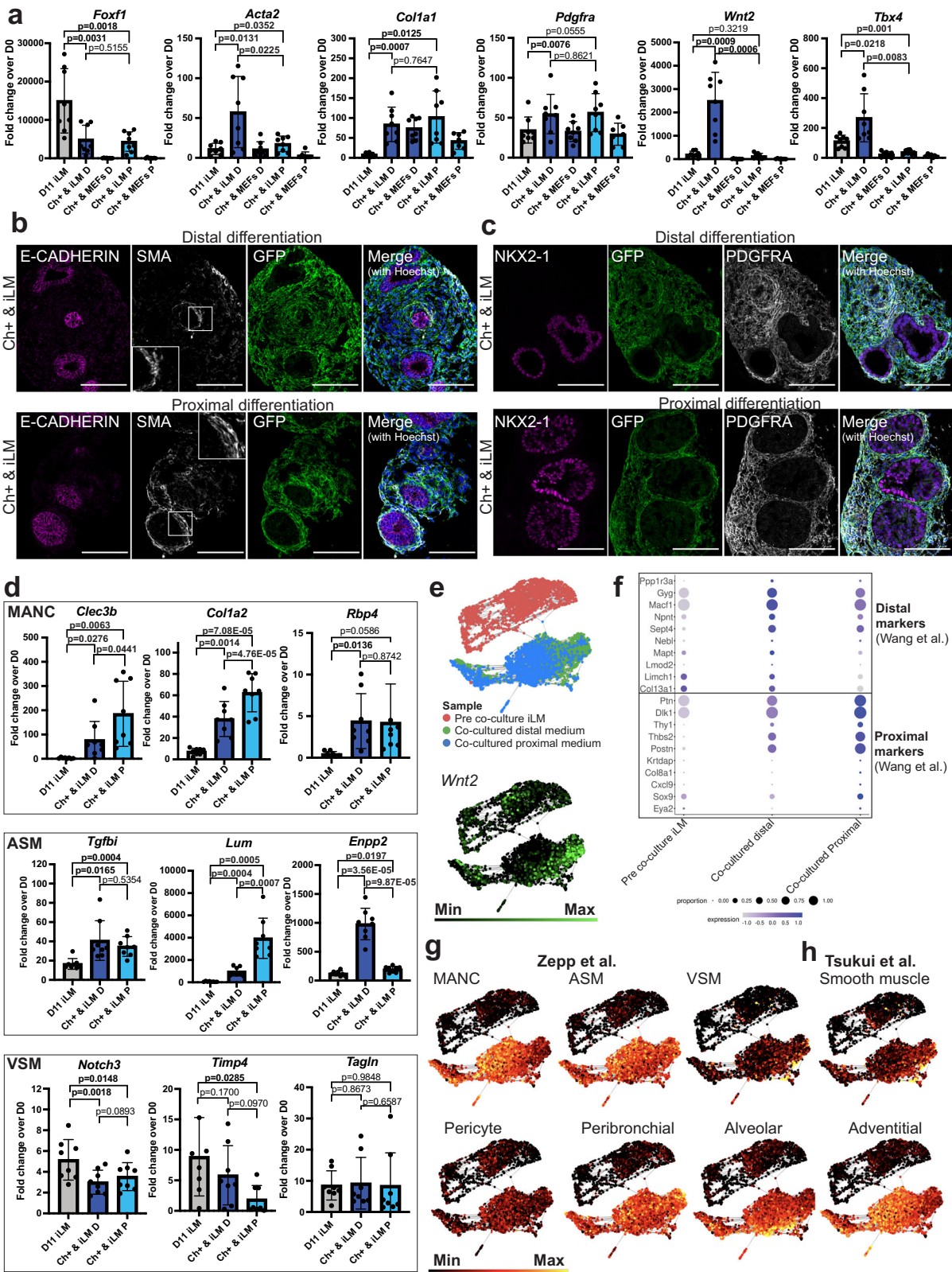

serum-free culture conditions for the derivation of this lineage. Moreover, this reporter/tracer facilitated the quantitative tracking and purification of the derived lung-specific mesenchymal progenitors as they emerged from mesodermal precursors, further allowing their prospective isolation, functional characterization, and transcriptomic profiling at single cell resolution. Our findings significantly extend prior reports of the derivation of developing mesenchyme enriched for lung-like features[2,9,29,50]. These prior studies relied on the developing embryo as an in vivo roadmap to guide the stepwise directed differentiation in vitro of iPSCs through mesodermal precursors into specified primitive mesenchymal progenitors, some of which expressed lung-like programs.

In contrast to developing lung epithelia for which there are now a variety of validated lineage specific reporters and marker genes,

**Fig. 6 | Cross-talk between induced lung mesenchyme and epithelium impacts mesenchymal differentiation in recombinant co-cultures. a** RT-qPCR data showing fold change expression relative to day 0 iPSCs of lung mesenchymal markers in day 11 iLM cells before co-culture (gray), and in iLM and MEFs after co-culture in distal (dark blue) and proximal (light blue) medium. Cells were re-sorted for GFP+ (iLM) or Cherry−/EPCAM− (MEFs) after 7 days of co-culture. N = 8. **b** Immunofluorescence images showing expression of GFP, E-CADHERIN and Smooth muscle actin (SMA) in Nkx2-1^mCherry+ cells co-cultured with iLM in distal or proximal medium for 1 week. Scale bars = 100 μm. **c** Immunofluorescence images showing expression of GFP, NKX2-1 and PDGFRA in Nkx2-1^mCherry+ cells co-cultured with iLM in distal or proximal medium for 1 week. Scale bars = 100 μm. **d** RT-qPCR data showing fold change expression relative to day 0 iPSCs of MANC, ASM and VSM markers in day 11 iLM cells before co-culture (gray), and in iLM after co-culture in distal (dark blue) and proximal (light blue) medium. N = 8. **e** SPRING plot showing sample identity and expression of *Wnt2* for single cell RNA-seq of iLM before and after distal and proximal co-culture. **f** Dot plot showing expression of 10 distal and 10 proximal markers in iLM before and after distal and proximal co-culture. These markers were previously found to be enriched in the distal versus proximal adult mouse lung by Wang et al., 2018. **g** SPRING plot showing expression of gene sets (top50 expressed genes) of adult lung mesenchymal cell types described by Zepp et al., 2021. **h** SPRING plot showing expression of gene sets (top50 expressed genes) of adult lung mesenchymal cell types described by Tsukui et al., 2020. All bars show mean ± sd. P values were determined by paired, two-tailed Student's t test. Significant (p < 0.05) p values are highlighted in bold font.

developing tissue-specific mesenchymal lineages remain poorly understood, due in part to a paucity of markers available for identifying or purifying tissue-specific mesenchymal cells. While a variety of mesenchymal markers selectively distinguish mesenchymal subsets, such as fibroblasts, myofibroblasts, smooth muscle, vascular endothelium, pericytes, adipocytes, and cartilage, most of these markers tend to be expressed across a variety of organs, and little is known about the tissue-specific differences between mesenchymal subsets across different organs, both during mesenchymal development and in the post-natal organism. We observed a variety of mesenchymal marker transcripts and gene elements expressed in our iPSC-derived mesenchyme that are also enriched in primary developing lung mesenchyme, including the Tbx4 enhancer element known to selectively identify developing lung-specific mesenchyme.

Importantly, while we did observe activation of our lung mesenchyme-specific reporter and expression of many lung mesenchymal markers in our iPSC-derived lung mesenchyme, we also detected some transcriptional differences between our iPSC-derived and primary lung mesenchymal cells, suggesting that while our iPSC-derived lung mesenchymal cells share similarities with primary lung mesenchyme, they are not equal. Lung mesenchymal markers *Tbx4* and *Wnt2* were expressed at significantly lower levels compared to primary cells. We also found that iPSC-derived lung mesenchyme clustered separately from primary cells, which could in part be caused by differences in expression of lung mesenchymal marker genes, but could also partially be due to cell culture effects unrelated to the lung-specific identity of our cells. In order to demonstrate that, despite some transcriptional differences between iPSC-derived and primary lung mesenchymal cells, our iPSC-derived lung mesenchyme can function as lung mesenchyme we employed a variety of functional readouts and concluded that our derived mesenchyme appeared to be lung-like. First, in contrast to developing non-lung embryonic mesenchymal cells, such as MEFs, our putative lung mesenchymal progenitors augmented epithelial lung lineage specification of co-cultured foregut endodermal cells and modulated the airway and alveolar differentiation programs of their epithelial descendants. Second, Tbx4-LER^GFP+ mesenchyme after recombination with developing lung epithelia, formed spatially organized organoids with mesenchymal descendants that were closely juxtaposed to these epithelia, and expressed a variety of upregulated maturing mesenchymal programs. Most importantly, after recombination gene expression changes in both mesenchymal and epithelial lineages provided evidence of bidirectional cross-talk, a key feature of the developing lung.

Our work adds to an emerging literature focused on deriving tissue-specific mesenchymal lineages from iPSCs. In many prior studies the technique of "co-development" has been employed, where both mesenchymal and epithelial lineages are simultaneously differentiated in heterogeneous cultures that take advantage of the known inefficiencies in directed differentiation of endodermal or mesodermal germ layers[17,30]. This approach has led, in some cases, to the formation of 3-dimensional structures containing both epithelial and mesenchymal lineages in close apposition, although whether the resulting mesenchyme was tissue-specific was not addressed[17,30]. We also attempted simultaneous co-development of mesoderm and lung endoderm, but found little evidence of lung-specific mesenchyme arising within the mesenchymal lineages that emerged in our culture conditions. In contrast, the separate derivation of putative lung-specific mesenchyme was more efficiently achieved through a dedicated mesodermal directed differentiation approach that we then optimized for the activation of the lung-mesenchymal specific reporter/tracer element through combinatorial stimulation of RA and Hh signaling. Our findings, however, only suggest that our co-development culture conditions were not optimal for lung specific mesenchymal specification compared to our dedicated mesodermal approach in our hands, and need not dissuade parallel efforts by others focused on co-development approaches for producing tissue-specific mesenchyme.

Our approach of separate derivation of epithelial and mesenchymal progenitors and subsequent sorting before recombinant co-culture resulted in lung-specific identity and purity of both lineages. This approach builds on a nascent literature using separate derivation of different germ layers to engineer functional multi-lineage organoids. For example, it has been shown that functional gastrointestinal organoids can be engineered using three germ layers separately derived from iPSCs[51]. In the lung, a recent study by Tamai et al. demonstrated that iPSC-derived (non-tissue specific) mesenchyme can support the development of separately generated iPSC-derived lung epithelium[50].

While iLM engineered through our protocol upregulated the expression of several sets of markers for mature lung mesenchymal cell types upon co-culture in distal or proximal medium, our results do not indicate that these cells are fully mature, and further work will be needed to generate progressively more mature lung mesenchymal sub-lineages, just as has been accomplished by iterative improvements in protocols for engineered lung epithelial lineages in the past 2 decades.

In conclusion, we have developed an in vitro iPSC-based model system for the derivation and study of early lung-specific mesenchyme with potential benefit for understanding basic mechanisms regulating tissue-specific mesenchymal fate decisions and future applications for regenerative medicine.

While engineered lung mesenchyme expressed many early lung mesenchymal markers at similar levels as primary embryonic lung mesenchyme, we did detect some differences between our engineered cells and primary controls. In particular, expression of the important lung mesenchymal marker *Wnt2* and downstream targets of the Wnt signaling pathway were expressed at lower levels in engineered lung mesenchymal cells. Interestingly, expression of *Wnt2* significantly increased upon co-culture with lung epithelial progenitors in distal differentiation medium. This suggests that lung epithelial progenitors provide signals necessary to induce *Wnt2* expression and potentially re-enforce and improve the lung-specific identity of engineered lung mesenchymal cells. Further work will be needed to identify these signals and modify the lung mesenchyme differentiation medium

accordingly in order to yield lung mesenchymal progenitors that resemble the primary control even more closely.

We have used mouse iPSCs to generate lung mesenchymal progenitors, harnessing a lung mesenchyme-specific reporter system. In order to study human lung mesenchyme development and to model human respiratory diseases involving the mesenchymal lineage, such as pulmonary fibrosis, establishing a protocol for the differentiation of human iPSCs into lung mesenchyme will be crucial. The *Tbx4* lung enhancer region is highly conserved in mouse and humans, however, it is unclear if this enhancer activates in the same lung-mesenchyme specific manner as in the mouse. Future work can focus on testing a human reporter and adapting our approach for the derivation of human lung-specific mesenchyme from patient-specific iPSCs.

## Methods

### Animal maintenance and isolation of E12.5 mouse embryonic lung mesenchyme

All experiments involving mice were approved by the Institutional Animal Care and Use Committee of Boston University School of Medicine. For the isolation of primary mouse embryonic tissue (C57BL/6 J strain, Jackson Laboratory #000664) on embryonic day (E) 12.5 timed-pregnant female mice were euthanized and embryonic lungs were isolated using Tungsten needles (Fine Science Tools, #10130-10) and kept in HBSS with 10% FBS on ice. Lungs were subsequently single cell digested using TrypLE (Gibco, #12640-013) for 15 min at 37 °C. Cells were treated with Red Blood Cell Lysing Buffer (Sigma, #R7767) and incubated with an BV421-conjugated anti-CD326 antibody (BD Biosciences, # 563214, 1/500 dilution) for 30 min on ice. Lung mesenchyme and epithelium were isolated by fluorescence activated cell sorting (FACS) for Epcam- versus Epcam+ cells. For cocultures of iLM with primary E12.5 embryonic lung epithelium and co-cultures of ESC-derived lung epithelium with primary lung mesenchyme, a previously published Nkx2-1[GFP] mouse line was used[42]. This mouse line has a C57BL/6 J background, was previously generated in our lab and is commercially available (Jackson Laboratory, #066764-JAX) Embryonic lungs were isolated and epithelial and mesenchymal cells were separated based on Nkx2-1[GFP]±. For co-cultures of iLM with adult alveolar epithelial type 2 (AT2) cells, primary adult AT2 cells were isolated based on flow cytometry sorting tdTomato+ lineage traced (Sftpc[TomatTr]) cells from single cell suspensions prepared from the lungs of adult (3–6 month old) mice carrying knock-in Sftpc-CreER[T2] and Rosa26-tdTomato alleles[44]. To generate these mice, Sftpc-CreER[T2] males (Jackson Laboratory, #028054, C57BL/6NCr background) were crossed with Rosa26-tdTomato females (Jackson Laboratory, #007914, C57BL/6 J background) and 3–6 months old Sftpc-CreER[T2]/Rosa26-tdTomato offspring were injected intraperitoneally with four doses of tamoxifen (20 mg/ml at 0.2 mg/g mouse weight, Thermofisher #J63509.03) every other day to induce Tomato expression in AT2 cells. Animals were harvested 3 weeks post tamoxifen administration. For control primary-primary co-cultures, a Pdgfra[GFP] reporter mouse line was used[52] (Jackson Laboratory, #007669, C57BL/6 J background). For isolation of adult Acta2+ lung cells, a previously published Acta2hrGFP/Cspg4-Cre-LSL-dsRed bifluorescent mouse line was used[45] and GFP+ live cells were sorted from lung digested single cell suspensions for scRNA-seq profiling. The Acta2hrGFP/Cspg4-Cre-LSL-dsRed has a C57BL/6 J background and was a generous gift from the lab of Alan Fine (Boston University).

### Generation and maintenance of mouse iPSC/ESC lines

A Tbx4-LER reporter/tracer iPSC line was generated by reprogramming tail tip fibroblasts from adult Tbx4-LER triple transgenic mice carrying an rtTA/LacZ encoding cassette under regulatory control of a Tbx4 lung-specific enhancer, a Tet-responsive Cre recombinase, and a fluorescent mTmG cassette (Tbx4-rtTA; TetO-Cre; mTmG, hereafter "Tbx4-LER")[35]. The Tbx4-LER triple transgenic mouse line was

generated in the lab of Wei Shi (University of Southern California) and has a C57BL/6 J background. Reprogramming was performed as previously published[24] using an excisable lentiviral STEMCCA-Frt vector followed by vector excision with transient flipase expression (adeno-flp)[24] as shown in Supplementary Fig. 1. A normal karyotype was documented by G-banding analysis (Cell Line Genetics). For lung epithelial differentiation, a Nkx2-1[mCherry] reporter ESC line[41] (generous gift from Dr. Janet Rossant, SickKids Research Institute, Toronto, Canada) was differentiated using protocols we have previously published for this line[27]. All iPSC/ESC lines were maintained on mouse embryonic fibroblasts in medium containing DMEM with 15% ESC-qualified fetal bovine serum (ThermoFisher, 16141061), Glutamax (Life Technologies, 35050-061), 0.1 mM Beta-mercaptoethanol (Fisher Scientific, #21985-023), Primocin (Fisher Scientific, NC9392943) and Leukemia inhibitory factor.

### Directed differentiation of iPSCs into lung mesenchyme

Tbx4-LER iPSCs were first differentiated into lateral plate mesoderm as previously published[36]. In brief, iPSCs were trypsinized and subsequently MEF-depleted by incubating in 10 cm tissue culture dishes (Fisher Scientific, #FB012924) in ESC medium at 37 °C for 30 mins to let MEFs attach. Remaining iPSCs in medium were collected by centrifugation at $300 \times g$ for 5 min. $1.5 \times 10^6$ cells were plated in 10 cm non-adherent Petri dishes (Fisher Scientific, #FB0875712) in complete serum-free differentiation medium (cSFDM). 500 ml cSFDM was composed of 375 ml IMDM (Thermo-Fisher, #12440053), 125 ml Ham's F12 (Cellgro, #10-080-CV), 5 ml Glutamax 100x (Invitrogen, #35050-061), 5 ml B27 supplement without vitamin A (Invitrogen, #12587-010), 2.5 ml N2 (Invitrogen, #17502-048), 3.3 ml 7.5% BSA (Life Technologies, #15260-037), 19.5 µL 1-thioglycerol (final concentration $4.5 \times 10$-4 M, Sigma, #M6145), 500 µl ascorbic acid (final concentration 50 µg/ml, Sigma, #A4544), and 1 ml Primocin (final concentration 100 µg/ml, Fisher Scientific, #NC9392943). Cells were incubated at 37 °C for 48 h to allow embryoid bodies to form. Embryoid bodies were then collected by gravity, re-plated and incubated in 10 cm non-adherent Petri dishes in cSFDM containing 2 ng/ml rhActivin A (R&D Systems, #338-AC), 3 ng/ml rhBMP4 (R&D Systems, #314-BP), and 3 ng/ml rmWNT3a (R&D Systems, #1324-WN-010) for 72 h. Day 5 mesodermal progenitors were then collected by gravity. For assessment of mesodermal differentiation efficiency, a small subset of embryoid bodies was removed, single cell dispersed by 0.05% trypsin, and incubated in anti-FLK1 (i.e., anti KDR) antibody (anti-FLK1; BD Biosciences, #560070, 1/50 dilution) or Isotype (BD Biosciences, # 553932, 1/50 dilution) for 30 min on ice. Cells were then stained with live/dead Calcein Blue (Life Technologies, #C1429) and analyzed by flow cytometry using a Stratedigm flow cytometer. For subsequent differentiation into lung mesenchyme, the remaining mesodermal progenitors were plated without enzymatic digestion on gelatin-coated (Stemcell Technologies, #07903) 12-well plates at an approximate cell density of 40,000 cells/cm² in cSFDM with 2 µM Retinoic acid (Sigma, #R2625). From day 7 to 9 of differentiation cells were cultured in cSFDM containing 2 µM Retinoic acid and 0.5 µg/ml purmorphamine (Stemcell Technologies, #72202), and then subsequently in cSFDM with 2 µM Retinoic acid until day 11–13 of differentiation. For the initial media screen (See Fig. 1e), 3 ng/ml or 200 ng/ml rmWNT3a (R&D Systems, #1324-WN-010) and/or 3 ng/ml rhBMP4 (R&D Systems, #314-BP) was also added to the medium from day 5 to 13, but rmWNT3A and rhBMP4 were not included in the day 5–13 medium in all subsequent experiments. 2 µg/ml Doxycycline (Sigma, #D3072) was added to the medium from day 5 to 13, or day 9–11, as indicated in the main text and figure legends. For analysis/sorting, cells were single cell dispersed by 0.05% trypsin and stained with live/dead Calcein Blue. Tbx4-LER[GFP] percentage was quantified using a Stratedigm flow cytometer. Sorting was

performed at the BU Flow Cytometry Core Facility using a MoFlo Astrios sorter.

## Directed differentiation of lung mesenchyme into more mature mesenchymal lineages

For smooth muscle differentiation, day 13 GFP + /cells were sorted and plated on gelatin-coated 12-well plates at a density of 40,000 cells/cm² and cultured in our previously published smooth muscle differentiation medium[36], consisting of cSFDM with 10 ng/ml rhPDGF-BB (Life Technologies, #PHG0046), 10 ng/mL rmTGF-β (R&D Systems, #7666-MB-005), and 10 ng/mL rhFGF2 (R&D Systems, #233-FB) for 4 days. Cells were harvested for RNA extraction, and differentiation efficiency was evaluated by RT-qPCR for the smooth muscle markers *Acta2* and *Tagln*.

For adipogenic differentiation, day 13 GFP+ cells were sorted, plated on gelatin-coated glass chamber slides (Millipore, #PEZGS0816) at a density of 40,000 cells/cm² and cultured in MesenCult Adipogenic differentiation medium (Stemcell Technologies, #05507), or cSFDM containing 2 µM Retinoic acid (control) for 9 days. Cells were then fixed in 4% paraformaldehyde for 30 min at room temperature and stained with hematoxylin and Oil Red O (Sigma, #O0625) to visualize nuclei and lipid droplets, respectively. Chambers were subsequently removed, slides were mounted and imaged using a Keyence BZ-X700 fluorescence microscope.

For endothelial differentiation, day 5 lateral plate mesoderm or day 7 differentiating lung mesenchyme was transferred to cSFDM supplemented with 100 ng/ml rmVEGF (R&D Systems, # 493-MV-005). For evaluation of differentiation efficiency, cells were dispersed into a single cell suspension using 0.05% trypsin and subsequently incubated with an anti CD144 antibody (anti-VE-Cadherin; BD Biosciences, #562242, 1/50) or isotype control (BD Biosciences #557690, 1/50) for 30 min on ice. Cells were stained with the live/dead Calcein Blue and analyzed using a Stratedigm flow cytometer.

## Directed differentiation of iPSCs/ESCs into lung epithelial progenitors

For lung epithelial differentiation, iPSCs/ESCs were MEF-depleted and differentiated according to our previously published serum-free protocol[27,38]. On day 13, cells were harvested by incubating in 2 mg/ ml Dispase (Gibco, #17105-041) for 1.5 h at 37 °C, 2× centrifugation for 2 min at 50 x *g*, and subsequent single cell dispersion by 0.05% trypsin. Cells were then stained for EPCAM using an anti-CD326 antibody (BD Biosciences, # 563214, 1/500) and the live/dead stain DRAQ7 (Biolegend, #424001, 1/100 dilution). Sorting for EPCAM + , Nkx2-1^mCherry+ or GFP+ cells, as indicated in the text, was performed at the BU Flow Cytometry Core Facility using a MoFlo sorter.

## Co-culture of engineered lung epithelial and mesenchymal cells

For epithelial-mesenchymal co-cultures, 24- or 48 well tissue culture plates were coated with a layer of 3D Matrigel (Corning, #354230) and incubated at 37 °C for 20 min to let the Matrigel solidify. Day 13 Nkx2-1^mCherry± cells were then combined with sorted day 11 Tbx4-LER^GFP+ cells or MEFs (control) at a 1/20 epithelial/mesenchymal cell ratio and seeded in the Matrigel coated plates at a density of 110 000 cells/cm². For distal epithelial differentiation, cells were incubated in cSFDM (with RA-containing B27 supplement) with 200 ng/ml rmWNT3A (R&D Systems, #1324-WN-010), 100 ng/ml rhFGF2 (R&D Systems, #233-FB), 100 ng/ml rhFGF10 (R&D Systems, # 345-FG), and 10 µM Y-27632 (Tocris, #1254). For proximal epithelial differentiation, cells were incubated in cSFDM (with RA-containing B27 supplement) with 100 ng/ml rhFGF2 (R&D Systems, #233-FB), 100 ng/ml rhFGF10 (R&D Systems, # 345-FG), 100 ng/ml Heparin Salt (Millipore Sigma, #H3393), and 10 µM Y-27632 (Tocris, #1254). In the distal differentiation medium, Y-27632 was removed after 48 h. Distal and proximal differentiation media were exchanged every 48 h for 7 days. Day 7 co-

cultures were imaged using a Keyence BZ-X700 fluorescence microscope. In order to re-separate co-cultured cells for sorting and RT-qPCR analysis, Matrigel was disrupted by incubating in 2 mg/ml Dispase for 1.5 h at 37 °C. Cells were then harvested by centrifugation at 300 x *g* for 5 min, single cell dispersed by 0.05% trypsin, and stained for Epcam using an anti-CD326 antibody (BD Biosciences, # 563214, 1/500 dilution) and the live/dead stain DRAQ7 (Biolegend, #424001, 1/100 dilution). Sorting was performed at the BU Flow Cytometry Core Facility using a MoFlo Astrios sorter. Upon gating for single and live cells, Nkx2-1^mCherry+ cells were isolated by further sorting for Epcam +/ mCherry+ cells, Tbx4-LER^GFP+ cells were isolated by sorting for Epcam−/GFP+ cells, and MEFs were isolated by sorting for Epcam-/ mCherry− cells. Harvested cells were resuspended in Qiazol Lysis reagent (Qiagen, #79306) and stored at −80 °C for further analysis. For co-cultures of ESC-derived day 13 Nkx2-1^mCherry+ epithelial cells and primary lung mesenchyme E12.5 lungs from the Nkx2-1^GFP mouse line[42] were isolated and single cell digested as described above. Primary lung mesenchyme was isolated by sorting for Nkx2-1^GFP- cells and epithelial and mesenchymal cells were combined at a 1/20 epithelial/mesenchymal ratio, plated on a bed of 3D Matrigel and co-cultured in distal medium for 1 week. For co-cultures of primary lung epithelium with day 11 Tbx4-LER^GFP+ cells, Nkx2-1^GFP+ cells from E12.5 lungs were isolated, combined with day 11 Tbx4-LER^GFP+ cells at a 1/20 epithelial/ mesenchymal ratio and co-cultured in distal medium for 1 week. For co-cultures with primary AT2 cells, Sftpc^TomatoTr+ lineage traced cells were isolated from 3 to 6 month old adult mouse lungs carrying knock-in Sftpc-CreER^T2 and Rosa26-tdTomato alleles[44] and combined with day 11 Tbx4-LER^GFP+ or freshly isolated Pdgfra^GFP+ primary lung mesenchymal cells at a 1/20 epithelial/mesenchymal ratio and plated on Matrigel coated plates in distal medium.

## Reverse transcriptase quantitative real time polymerase chain reaction (RT-qPCR)

RNA was extracted using an RNeasy Plus Mini Kit (QIAGEN, #74136) according to the manufacturer's instructions. RNA was eluted in 30 µl nuclease-free water and quantified using a NanoDrop ND-1000 microvolume spectrophotometer (ThermoFisher Scientific, Waltham, MA). cDNA was generated from 150 ng of RNA in a 20 µl reaction using a High-Capacity cDNA Reverse Transcription Kit (Applied Biosystems, #4368814). qPCR was performed using TaqMan Fast Universal PCR Master Mix (ThermoFisher, #364103). cDNA was diluted 1/16 and 4 µl of diluted cDNA was used in a 10 µl qPCR reaction, using an Applied Biosystems QuantStudio7 384-well System. Relative expression was calculated using the the the $2^{(-\Delta\Delta Ct)}$ method[53], using 18 S rRNA (Thermo-Fisher, #4318839) as the internal reference gene and undifferentiated (day 0) iPSCs as the reference sample. Undetected Ct values were set to the maximum number of cycles (40) to allow fold change calculations. TaqMan gene expression arrays were purchased from ThermoFisher (Supplementary Data 7).

## Embedding of organoids and immunofluorescence experiments

In order to allow for fixation and paraffin embedding, lung epithelial-mesenchymal co-cultures were plated on glass chamber slides (Millipore, #PEZGS0816) coated with 3D Matrigel. On day 7 of co-culture, medium was removed from the chambers and another layer of 3D Matrigel was added on top of the co-cultures and allowed to solidify at 37 °C for 20 min. Chambers were removed from the slides and Matrigel pellets containing lung epithelial-mesenchymal organoids were carefully removed, transferred to 12-well plates, and incubated in 4% paraformaldehyde for 1 h at room temperature. Samples were washed, dehydrated, and paraffin-embedded. 5 µm sections were de-paraffinized and treated with citric acid-based Antigen Unmasking Solution (Vector Laboratories, #H-3300) according to the manufacturer's instructions. After washing in PBS and blocking with 10% Normal Donkey Serum

(Sigma, #D9663), 2% BSA (Fisher Scientific, #BP1600), and 0.5% TritonX-100 (Sigma) for 1 h at room temperature, sections were incubated in primary antibody (chicken anti-GFP, Aves labs GFP-1010, 1/500; goat anti-PDGFRA, R&D Systems AF1062, 1/200; rat anti-SOX2, ThermoFisher 14-9811-82, 1/250; rat anti-E-cadherin, Invitrogen 13–1900, 1/200; rabbit anti-SMA, Abcam ab5694, 1/500; rabbit anti-proSFTPC, Abcam ab211326, 1/250; rabbit anti-SOX9, Abcam ab185966, 1/500; rabbit anti-TTF1 (Nkx2-1), Abcam ab76013, 1/500; rabbit anti-p63, Cell Signaling 13109, 1/500, goat anti-RFP, MyBioSource, MBS448122, 1/500) resuspended in ¼ blocking buffer over night at 4 °C, washed 3 × with PBS, and incubated in secondary antibody (Alexa Fluor® 488 AffiniPure Donkey Anti-Chicken IgY (IgG) (H + L), Jackson ImmunoResearch 703-545-155, 1/500; Alexa Fluor® 647 AffiniPure Donkey Anti-Rabbit IgG (H + L), Jackson ImmunoResearch 711-605-152, 1/500; Cy"3 AffiniPure Donkey Anti-Goat IgG (H + L), Jackson ImmunoResearch 705-165-147, 1/500; Cy"3 AffiniPure Donkey Anti-Rat IgG (H + L), Jackson ImmunoResearch 712-165-153, 1/500) and Hoechst 33342 (ThermoFisher, # H3570, 1/500) for 2 h at room temperature. Slides were subsequently washed 1× with PBS and mounted using FluorSave mounting reagent (Millipore, # 345789). Slides were imaged using a Zeiss LSM710 confocal microscope.

## Single cell RNA sequencing (scRNA-seq)

For scRNA-seq of iPSC-derived and primary lung mesenchymal cells, Tbx4-LER iPSCs were differentiated into lung mesenchyme using the directed differentiation protocol described above, and subsequently sorted for Tbx4-LER[GFP]+ and Tbx4-LER[GFP]- cells. In parallel, to generate co-developed lung mesenchyme Tbx4-LER iPSCs were differentiated using the lung epithelium protocol as described above, stained with EPCAM antibody (anti-CD326, BD Biosciences, # 563214, 1/500), and sorted for Tbx4-LER[GFP] + (cLM) cells and EPCAM + /Tbx4-LER[GFP]− (i.e., epithelial) cells. cLM and epithelial cells were mixed at a 3:1 ratio. For primary controls, primary embryonic lungs from embryonic day E12.5 were isolated and single cell digested as described above, stained with EPCAM antibody (anti-CD326 antibody, BD Biosciences, # 563214, 1/500), and sorted for EPCAM- and EPCAM+ cells. EPCAM- and EPCAM + cells were mixed at a 3:1 ratio. For the single cell RNA-seq of adult mouse Acta2+ cells, GFP+ live cells were sorted from adult Acta2hrGFP/Cspg4-Cre-LSL-dsRed mice[45]. For the single cell RNA-seq of co-cultured iLM, Nkx2-1[mCherry]+ epithelial progenitors were cultured in distal or proximal medium either alone or with iLM and re-sorted based on EPCAM-/Tbx4-LER[GFP] + (iLM) and EPCAM + /Nkx2-1[mCherry] + (epithelial cells). iLM and epithelial cells were then mixed at a 1:1 ratio for single cell capture. In parallel, pre co-cultured iLM and Nkx2-1[mCherry]+ epithelial progenitors were also submitted for sequencing. All sorts were performed at the BU Flow Cytometry Core Facility using a MoFlo sorter and cells were resuspended in FACS buffer at a concentration of 1000 cells/µl. Single cell capture (10× Chromium instrument; 10× Genomics, Pleasanton, CA), library preparation (Chromium Single-Cell 30 Library Kit; 10× Genomics), and sequencing was performed at the Boston University Single Cell Sequencing Core Facility. Sequencing libraries were loaded on an Illumina NextSeq500 (scRNA-seq of primary and iPSC-derived lung mesenchyme, and scRNA-seq of adult mouse Acta2+ cells) or NextSeq2000 (scRNA-seq of co-cultured iLM) with a custom sequencing setting to obtain a sequencing depth of ~40k reads/cell. Sequencing files were mapped to the mouse genome reference (GRCm38) supplemented with GFP, tdTomato, and mCherry sequences using CellRanger v3.0.2. After inspection of the quality control metrics, cells with 15–35% of mitochondrial content and <800 detected genes were excluded for downstream analyses. In addition, doublets were also excluded for downstream analysis. The average sample had a mean of 56,451 reads per cell (ranging from 49,239 to 61,912 depending on the sample.). The median number of genes detected per cell on the average sample was 3793 genes. We normalized and scaled the unique molecular identifier (UMI) counts using the regularized negative binomial regression (SCTransform v0.3.5)[54]. Following the standard procedure in Seurat's pipeline (v4.1.0), we performed linear dimensionality reduction (principal component analysis) and used the top 40 principal components to compute the unsupervised Uniform Manifold Approximation and Projection (UMAP)[55]. Further non-linear dimensionality reduction was performed with SPRING[56]. For clustering of the cells, we used the Louvain algorithm (v0.4.3)[57] which was computed at a range of resolutions from 1.5 to 0.05 (more to fewer clusters). Cell cycle scores and classifications were done using the Seurat's cell-cycle scoring and regression method[58]. Cluster specific genes were calculated using MAST (v1.20.1) framework in Seurat wrapper[59]. For computational analysis of iPSC-derived Tbx4-LER[GFP]± versus primary lung mesenchyme, Louvain clustering with a resolution of 0.1 was used, and epithelial and other non-mesenchymal (i.e., endothelial, immune cell) clusters were removed for further analysis. We used MAST framework in Seurat wrapper to identify the top 50 DEGs in each cluster of interest, represented as heatmaps (Supplementary Fig. 2b, 3a). For computational analysis of single cell RNA-seq of adult Acta2+ lung cells, Louvain clustering with a resolution of 0.25 was used, and clusters were annotated as MANCs, ASM and VSM based on enrichment of gene sets identified by Zepp et al, 2021[46] (detailed in Supplementary Data 3). An additional minor cell cluster was presumed to be non-mesenchymal hematopoietic cells based on expression of the pan-hematopoietic marker, *Ptprc* (CD45). For computational analysis of the single cell RNAseq of co-cultured iLM, all Epcam+ clusters were excluded and further analysis was focused on mesenchymal clusters only. For combined UMAP plots (Supplementary Fig. 8) Seurat object obtained from Zepp et al., 2021[46] was transformed and normalized with our dataset using SCTransformtion. The cell type annotation was used as defined by Zepp et al., 2021.

## Statistics and reproducibility

All bar graphs represent mean and standard deviation and each data point represents one independent biological replicate. Number of biological replicates for RT-qPCR/flow cytometry analysis are indicated in the figure legends, at least 3 replicates were used for all experiments. Statistically significant differences between conditions were determined using one-tailed or two-tailed, unpaired or paired Student's *t* test, as indicated in the figure legends. Micrographs of live cell cultures (Fig. 3b, c, Fig. 5e, Supplementary Fig. 4b) and Alkaline phosphatase (Supplementary Fig. 1e), SMA (Supplementary Fig. 4d) and Oilred O (Supplementary Fig. 4e) stains show representative fields of view from whole wells. Each experiment was performed at least 3 times. Micrographs of fixed and sectioned co-cultures (Fig. 3d, Fig. 4b, Fig. 5b, h, i, Fig. 6b, c) show representative fields of view. Each staining was performed on one whole microscopy slide, which contained 10–20 recombinant organoids.

## Reporting summary

Further information on research design is available in the Nature Portfolio Reporting Summary linked to this article.

## Data availability

The scRNA-seq data reported in this publication have been deposited in NCBI's Gene Expression Omnibus. The scRNA-seq dataset comparing engineered to primary embryonic lung mesenchyme is available under accession code GSE203245. The scRNA-seq dataset focusing on mouse lung Acta2+ cells is available under accession code GSE203243. The scRNA-seq dataset focusing on co-cultured iLM is available under accession code GSE228313. GRC38m mouse genome [https://ftp.ncbi.nlm.nih.gov/genomes/all/GCF/000/001/635/GCF_000001635.27_GRCm39/GCF_000001635.27_GRCm39_genomic.fna.gz] was used as

reference genome for sequence mapping. Source data are provided with this paper.

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

## Acknowledgements

The authors wish to thank all members of the Kotton lab for insightful discussions. We thank Brian R Tilton from the Boston University Flow Cytometry Core Facility for cell sorting, Yuriy Alekseyev, Katerina Zhang, and Ashley LeClerc of the Boston University School of Medicine (BUSM) Single Cell Sequencing Core for RNA sequencing, and Hui Chen and Joanne Chiu for help with mouse work. This work was supported by the Swiss National Science Foundation (P2ELP3_191217 and P500PB_206631 to A.B.A.) and the U.S. National Institutes of health (U01HL134745, U01HL134766, U01HL152976, and R01HL095993 to D.N.K., R01HL141352 and R01HL146541 to W.S., and R56 DE028545-01 and R01HL158965-01 to L.I.). L.I. was also supported by University at Buffalo (UB) Research Foundation Start-up Funds. Schematics were created with BioRender.com.

## Author contributions

Conceptualization and Methodology, D.N.K., A.B.A., and H.A.M. Investigation and analysis, A.B.A., H.A.M., L.M., G.K., B.R.T., Y.L., and L.I. Computational analysis C.V-M., J.L.-V., P.B., and F.W., Resources, W.S. Visualization and Writing, A.B.A. and D.N.K. Supervision, D.N.K.

## Competing interests

The authors declare no competing interests.
