## [Peer Review File · Nature Communications]

Directed differentiation of mouse pluripotent stem cells into functional lung-specific mesenchymeREVIEWER COMMENTS

Reviewer #1 (Remarks to the Author):

In this manuscript, Alber et al claim that they have developed conditions to directed differentiation of mouse induced pluripotent stem cells (iPSCs) into lung mesenchymal cell populations (iLM). The authors first generated iPSCs from a triple transgenic mouse line that harbors a dox-inducible fluorescent reporter cassette under the regulatory control of Tbx4 lung specific enhancer element. Upon doxycycline administration, this mouse line (Tbx4-LER) permanently labels cells with a green fluorescent protein and thereby allows for identifying lung mesenchymal cells and their derivatives. The authors used iPSCs from this mouse line to find culture conditions that can induce their differentiation into lung mesenchymal cells. The authors used immunostaining, RT-PCR, and scRNA-seq analyses and find that retinoic acid (RA) and sonic hedgehog (Shh) signaling are critical for the induction of iLM in ex vivo cultures. These findings are in line with pathways that activate lung mesenchyme in developing lungs in vivo. The authors then tested the ability of lung mesenchymal (iLM) cells to support lung epithelial progenitors as well as their differentiation. The authors also claim that co-culturing with lung epithelial progenitors can enhance the survival and the fate of iLM cells in cultures.

In summary, this manuscript attempts to fill a major gap in directed differentiation conditions and provide a roadmap for future studies to utilize iLMs for developmental studies and disease mechanisms. Over the years, multiple groups have mainly focused on epithelial lineages and have almost ignored mesenchymal cells. While the

Major points:

1. Effect of growth factors and small molecules: The authors tested different factors, including RA, BMP4, WNT3A, and PMA and they removed one or two factors at a time in an attempt to find a minimal cocktail that supports lateral plate mesoderm cells to differentiate to lung mesenchyme. From Figure 1e, it is evident that removal of WNT3A showed similar effect as RA and even greater effect than removing PMA from the medium. However, combinatorial removal of BMP4 and WNT3A showed the opposite effect by promoting the maximal differentiation. Therefore, it is not clear what role Wnt stimulation plays in inducing iLM. What happens if you remove WNT3A while providing RA and PMA?
2. In this manuscript authors exclusively tested conditions for mouse iPSCs. Do the established conditions also work for human iPSCs?
3. In Figure 2g and in Supplementary Figure 2f, RT-qPCR detected no significant change in the expression of Wnt2b and Col1a1 or Acta2 between iLM and iLMNeg cells, respectively. This suggests that these two cell populations may not be that different. Authors should check for some additional genes listed by Han et al, 2020 to clarify this.
4. The presence of an iLMNeg cells with similar differentiation capacity to iLM cells is also evident from figure 3c. TBX4-LERGFP+ and TBX4-LERGFP- cells both can differentiate to smooth muscle cells and express Acta2 and Tagln markers significantly higher than the control and they both expressed Foxf1 with no detectable difference. This reviewer wonders whether the transgene constructs used here under represent actual Tbx4 expression? The authors could check this by immunostaining for Tbx4 on sorted TBX4-LERGFP+ and TBX4-LERGFP-. Additionally, the authors could check gene expression profiles to find potential differences between these two populations.
5. In Fig-5, the authors state that epithelial cells generate both proximal and distal airway cell types upon co-culture with iLMs in proximal or distal differentiation conditions, respectively. How about the iLM derivatives in proximal and distal differentiation medium conditions? Do they show markers of proximal airway or distal alveolar mesenchymal markers. The authors could use immunostaining or FISH for markers specific to lung mesenchymal cells in these respective regions to address this.

6. One key missing piece in this manuscript is whether iLMs can generate mature lung mesenchymal cells found in adult lungs? Although the authors showed expression of a couple of markers by RT-PCR or immunostaining, some of the markers tested in this manuscript do not distinguish distinct lung mesenchymal cell populations. The authors compared their scRNA-seq data with that of Zepp et al and claim that they could identify MANCs, ASM, and VSMS. While Zepp et al., have identified some sub-populations of mesenchymal cells, recent studies including Tsukui et al. have provided additional subsets of adult mouse lung mesenchymal cells and associated markers. These recent studies identified per-bronchiolar fibroblasts, adventitial fibroblasts, alveolar (interstitial) fibroblasts etc. Therefore, this reviewer suggest that the authors compare their data with Tsukui et al. Nature communications paper.

Reviewer #2 (Remarks to the Author):

Albert et al. established a three-step protocol to selectively induce the lung mesenchyme (LM) from mouse iPS cells harboring GFP under the control of a lung-specific Tbx4 enhancer. After analysis of the induced lung mesenchyme (iLM) by scRNA-seq, the authors combined the iLM with ES cell-derived lung epithelia to show the increased yields of epithelia in the organoids, along with increases in some mesenchymal genes. The combination of iLM with ES cell-derived endodermal cells also enhanced the yields of lung epithelial cells.

Although the approach of using LM-specific GFP is useful, the study does not strictly address similarities between the iLM and its in vivo counterpart, or the superiority of the iLM over the conventional lung mesenchyme co-induced with lung epithelia (cLM). Furthermore, the study does not include functional assays involving organoids generated by combining lung epithelia with cLM or LM in vivo. In the current form, it remains unclear how similar the iLM is to the LM in vivo; it also remains unclear whether the iLM is superior to the cLM.

Major comments

1. It remains unclear how similar the iLM is to its in vivo counterpart or whether the iLM is superior to the cLM. The authors should merge all scRNA-seq data (GFP-positive iLM, GFP-negative iLM, cLM, and E10.5&12.5 LM in vivo) using Seurat software; they should present UMAP plots that show the overlap of the iLM cluster, but not the other clusters, to the in vivo clusters. Theoretically, E12.5 LM should separate into multiple clusters, including immature progenitors, proximal and distal mesenchyme, smooth muscles, and many others. I would like the merged plots to be shown with better resolution. It would also be informative to perform hierarchical clustering analysis and/or calculate Pearson correlation coefficients to show quantitative similarities and differences between the compared samples. Similarly, organoids generated by combining iLM and lung epithelia should also be analyzed by scRNA-seq and merged in the UMAP plots described above.
2. However, I suspect that the iLM may not be similar to its in vivo counterpart, because Fig. 2 shows significantly lower expression levels of Tbx4 and Wnt2, while Wnt2b is de-repressed. To achieve significant overlaps in the UMAP plots, the expression levels of most key genes would need to reach the levels observed in vivo. The increased expression levels of Shh downstream genes are presumably caused by Shh treatment, which does not serve as evidence of similarity. Because Fig. 1e shows the essential role of Wnt for GFP expression, additional Wnt activation by Wnt3a or CHIR99021 may be helpful. It may also be helpful to optimize the concentrations of all reagents. It is important to compare the expression levels of multiple genes in the iLM to the levels in the in vivo counterpart at

multiple time points. GFP is a useful monitoring tool, but strict reliance on GFP percentages may provide misleading evidence concerning induction.

3. Another possible reason for the insufficient gene expression is that the proposed protocol does not accurately consider the corresponding embryonic stages *in vivo*. Under the assumption that ES cells represent the E3.5 blastocyst stage, I wonder whether it is reasonable to conclude that ES cells differentiated for 5 days and 13 days correspond to E8.5 and E12.5, respectively. In particular, STEP 3 lasted for 8 days with a fixed condition. Perhaps better quality can be achieved by shortening STEP 3 or subdividing it into multiple steps. Overall, the authors should clarify which stage they are targeting (e.g., E10.5, E12.5, or later).

4. Notably, *Wnt2* is upregulated in the iLM, upon combination with ES cell-derived lung epithelia; this finding suggests crosstalk between the mesenchyme and epithelia. This is also related to Comment 3. If the iLM corresponds to the stage when *Wnt2* remains low (e.g., E10.5), the crosstalk may upregulate *Wnt2* expression to the level observed at E12.5. Thus, it is essential to compare the expression levels of key genes in the iLM and cultured organoids to those expression levels within *in vivo* samples at various stages (e.g., E10.5 and 12.5), using qPCR and eventually scRNA-seq.

5. Although the authors used MEFs as negative controls for functional assays (organoid generation), they should use the cLM to demonstrate the advantages of their new induction protocol in the main experiments in Figs. 4–7. Together with scRNA-seq analysis (Comment 1), the findings would indicate whether the quality or yield of the induced cells have improved. It is also important to include the LM *in vivo* for organoid generation as an ideal reference to clarify the functional similarity of iLM to its *in vivo* counterpart. For example, Fig. 4 shows the disturbed differentiation of epithelial cells when combined with iLM, but it is unclear whether this occurs when combined with the LM *in vivo*. The authors should also explore the effects of exogenous cytokine omission on the organoids. Ref. 31 showed the dominant role of the LM in the fate determination and differentiation of lung epithelia. If this dominant role is a consistent phenomenon, the genuine mesenchyme should direct proximal or distal lung fate and promote epithelial differentiation by providing key signaling factors. However, the authors added exogenous cytokine cocktails to control such fates and the iLM appeared to antagonize such differentiation, rather than eliminating the need for mesenchyme-derived factors. The authors should demonstrate that the observed phenotypes represent physiological processes.

6. The authors sometimes compared iLM with adult LM, which is not valid. For example, Ref. 41 showed that AT2 cells can be maintained in the presence of adult LM without exogenous factors. However, the authors combined AT2 cells with iLM, then added various cytokines to show the survival of AT2 cells (Fig. 6). This approach does not prove that the functionality of iLM is similar to that of adult LM. Fig. 7 also shows analysis of the adult lung; the obtained markers were used for qPCR analysis of the iLM-containing organoids. To support the authors' claims, scRNA-seq analysis should be utilized to compare the differentiation/maturation states of the iLM-containing organoids with the adult lung or—more realistically—embryonic lung.

Minor comments

1. Fig. 3 simply shows that the iLM has generic mesenchymal stem cell-like properties; it does not strengthen the authors' claim that the iLM is similar to the LM *in vivo*.

2. The authors used ES cell-derived lung epithelial cells for organoid generation. However, because the quality of ES cell-derived cells is not completely equal to the quality of their *in vivo* counterparts, primary epithelial cells from transgenic mice may sometimes be more useful for rigorously testing the functional competence of the induced LM.

Reviewer #1 (Remarks to the Author):

This manuscript attempts to fill a major gap in directed differentiation conditions and provide a roadmap for future studies to utilize iLMs for developmental studies and disease mechanisms. Over the years, multiple groups have mainly focused on epithelial lineages and have almost ignored mesenchymal cells.

Major points:

1. Effect of growth factors and small molecules: The authors tested different factors, including RA, BMP4, WNT3A, and PMA and they removed one or two factors at a time in an attempt to find a minimal cocktail that supports lateral plate mesoderm cells to differentiate to lung mesenchyme. From Figure 1e, it is evident that removal of WNT3A showed similar effect as RA and even greater effect than removing PMA from the medium. However, combinatorial removal of BMP4 and WNT3A showed the opposite effect by promoting the maximal differentiation. Therefore, it is not clear what role Wnt stimulation plays in inducing iLM. What happens if you remove WNT3A while providing RA and PMA?

We thank the reviewer for this comment and have edited our manuscript accordingly to include new data to clarify the role of canonical Wnt signaling in our protocol, as we agree the results in Figure 1e can be confusing, since we did not formally test the requirement for Wnt signaling. As the reviewer correctly surmised, in Figure 1e removing WNT3A (but keeping BMP4) resulted in less GFP yield implying a possible role for WNT signaling in our protocol. However, supplemental WNT3A was dispensable in the final minimal media (see the condition labeled “minus BMP4” in Figure 1e, which represents WNT3A added to the minimal medium). While these results suggest that WNT3A *supplementation* is dispensable for lung mesenchymal lineage specification in our model, they do not exclude a role for Wnt pathway activation (e.g. from endogenous Wnts). **In response to the reviewer’s question and to further investigate the role of Wnt signaling in inducing iLM we have added an experiment showing the effect of Wnt inhibition using XAV as an inhibitor of canonical Wnt signaling**, while still providing RA and PMA (Figure 1f). We show that adding XAV to the medium significantly decreases the percentage of GFP+ cells, suggesting that successful lung mesenchymal differentiation requires active Wnt signaling (from endogenous signaling), but there is no need for additional exogenous Wnt supplementation as we did not find any dose response effects from adding low (3ng/ml) or high (200ng/ml) doses of WNT3A.

We have added our new experimental data (Figure 1f) and edited the text accordingly in response to the reviewer’s query.

2. In this manuscript authors exclusively tested conditions for mouse iPSCs. Do the established conditions also work for human iPSCs?

Yes, similar conditions work for human iPSCs. As a follow-up to our generation of induced lung mesenchyme from mouse iPSCs we have generated a human iPSC line carrying a reporter based

on the same lung mesenchyme-specific enhancer region of the *Tbx4* gene as the mouse reporter line and have established a directed differentiation protocol. While our human differentiation data is largely consistent with our mouse work, the details of these experiments are many and would require several main figures. Thus, we feel that including this data would be beyond the scope of this manuscript, due to the need to introduce and validate our newly engineered human reporter line, as well as the careful characterization of the human putative induced lung mesenchyme. We thus suggest to address the differentiation of human iPSCs using our lung mesenchyme-specific reporter in a separate manuscript. Although we do not include this new data in our revised manuscript, in response to the reviewer's query, we provide the following details: In our human iPSC reporter line, the *Tbx4* lung enhancer element directly drives the expression of tdTomato (see Figure A below). As with our mouse directed differentiation protocol our current human differentiation protocol includes differentiation into lateral plate mesoderm as published by Kwong et al., 2019¹, followed by subsequent differentiation into lung mesenchyme (Figure B below). Our data suggests that stimulation of Retinoic acid and Hh signaling in human iPSC-derived lateral plate mesoderm also promotes lung mesenchymal differentiation, as evidenced by the emergence of Tomato+ cells (~40% efficiency; see Figure C-D below).

A: Schematic of the human Tbx4-LERTomato reporter line. **B:** Human lung mesenchyme directed differentiation protocol. **C:** Flow cytometry showing Tbx4-LERTomato fluorescence compared to non-fluorescent wildtype control. **D:** Quantification of Tbx4-LERTomato fluorescence on day 30 of differentiation.

3. In Figure 2g and in Supplementary Figure 2f, RT-qPCR detected no significant change in the expression of *Wnt2b* and *Col1a1* or *Acta2* between ILM and ILMNeg cells, respectively. This suggests that these two cell populations may not be that different. Authors should check for some additional genes listed by Han et al, 2020 to clarify this.

We agree and thank the reviewer for this comment. We have attempted in our revision to better clarify both the gene similarities and differences between these two populations. We agree that

iLM and iLM^{Neg} populations, being both mesenchymal, express canonical mesenchymal markers (e.g. *Col1a1* and *Acta2*) at similar levels, but differ in expression of a subset of important markers (as shown in the heatmap in Supplementary Figure 2b, as well as the RT-qPCR in Figure 2h and Supplementary Figure 3d). We expect the two populations to share some transcriptomic overlap, as correctly surmised by the reviewer since the cells undergo the same differentiation protocol, pass through the same lateral plate mesodermal stage, and are exposed to the same growth factors. Therefore, even though the iLM^{Neg} cells do not activate the *Tbx4*-*LER^{GFP}* reporter, they are expected to still express many overlapping mesenchymal markers. In response to the reviewer's questions, we have attempted to clarify that both our RT-qPCR and scRNAseq analysis indicate that iLM^{Neg} cells are also classified as mesenchymal, consistent with the observed expression of more general mesenchymal markers such as *Col1a1* or *Acta2*. We thus think that the observed similarity between iLM and iLM^{Neg} cells is to be expected and we have modified the manuscript to better explain this observation.

In addition, to further investigate the differences between iLM and iLM^{Neg} cells, we have added a pairwise comparison to our single cell RNAseq analysis, showing the top50 differentially expressed genes between the two populations (Supplementary Figure 3a). This pairwise comparison shows an enrichment of target genes of the RA and Hh signaling pathways (such as *Cyp26b1* and *Ptch1*) in iLM cells compared to iLM^{Neg} cells, suggesting that iLM cells are more responsive to the RA and Hh-stimulating factors present in our culture medium. We also find an enrichment of markers of embryonic lung mesenchymal genes that are not direct targets of these signaling pathways, such as *Foxf1*. Importantly, several genes found to be enriched in early embryonic lung mesenchyme by Han et al., 2020² were among the top50 enriched genes in iLM in our dataset, such as *Sparcl1*, *Foxp2* and *Pde5a*. In contrast, in iLM^{Neg} cells we find an enrichment of mesothelial markers such as *Wt1* and *Upk3b*, as well as the embryonic esophageal marker *Wnt4*, suggesting that while iLM and iLM^{Neg} cells share many similarities, iLM^{Neg} cells are less lung-like. We have added paragraph in our manuscript to discuss these observations (line 248-260).

Finally, we have quantified the expression of additional genes listed by Han et al. by RT-qPCR, as suggested by the reviewer. We show that 3 of these genes (*Sparcl1*, *Pde5a*, and *Tnfrsf19*) are expressed at significantly higher levels in iLM compared to iLM^{Neg} cells, whereas the expression of these genes does not differ in iLM compared to primary lung mesenchymal cells (Figure 2i, Supplementary Figure 3c), in line with iLM cells being more lung mesenchyme-like than iLM^{Neg} cells.

4. The presence of iLM^{Neg} cells with similar differentiation capacity to iLM cells is also evident from figure 3c. TBX4-LER^{GFP}+ and TBX4-LER^{GFP}- cells both can differentiate to smooth muscle cells and express Acta2 and Tagln markers significantly higher than the control and they both expressed Foxf1 with no detectable difference. This reviewer wonders whether the transgene constructs used here under represent actual Tbx4 expression? The authors could check this by immunostaining for Tbx4 on sorted TBX4-LER^{GFP}+ and TBX4-LER^{GFP}-. Additionally, the authors could check gene expression profiles to find potential differences between these two populations.

We completely agree with the reviewer that both populations are mesenchymal and can similarly differentiate to general mesenchymal derivatives, such as smooth muscle as well as *Foxf1* or *Tbx4* expressing mesenchymal lineages, but differ in terms of lung specific mesenchymal competence. As discussed above, to better define the differences between iLM and iLM^{Neg} cells, we have added a pairwise comparison to our single cell RNAseq analysis, showing the top50 differentially expressed genes between the two populations (Supplementary Figure 3a). Also, as discussed in comment #3 above, we have edited our manuscript and added additional results to make it more clear that our data suggests that *Tbx4*-LER^{GFP}- cells are also mesenchymal, as stated by the reviewer. Since differentiation towards the smooth muscle lineage is not restricted to lung mesenchymal progenitors, it is expected that *Tbx4*-LER^{GFP}- cells are also competent to differentiate towards the smooth muscle lineage, and we have edited our manuscript to further underline this observation (see line 319-325). In response to the reviewer's question about *Tbx4*, we have also edited our revised manuscript to make more clear that *Tbx4* transcripts are detectable in both iLM and iLM^{Neg} populations (see line 257-260). This is expected because our reporter is based not on the *Tbx4* gene or promoter, but rather on a lung mesenchyme-specific *enhancer* element within the *Tbx4* locus. Thus, the reporter/tracer does not necessarily reflect *Tbx4* expression itself, and *Tbx4* can still be expressed in iLM^{Neg} cells (and indeed is expressed to some degree in both populations) due to activation of other (non-lung specific) enhancer elements. To further illustrate the expression of *Tbx4* in *Tbx4*-LER^{GFP+} vs *Tbx4*-LER^{GFP-} cells we include a Figure (Supplementary Figure 3b) showing the expression of *Tbx4* transcripts in these two populations, which is enriched in the iLM population, but is still detectable and expressed in the iLM^{Neg} population.

5. In Fig-5, the authors state that epithelial cells generate both proximal and distal airway cell types upon co-culture with iLMs in proximal or distal differentiation conditions, respectively. How about the iLM derivatives in proximal and distal differentiation medium conditions? Do they show markers of proximal airway or distal alveolar mesenchymal markers. The authors could use immunostaining or FISH for markers specific to lung mesenchymal cells in these respective regions to address this.

We thank the reviewer for this suggestion and we agree. To address this question, we have now performed a single cell RNAseq experiment, analyzing iLM before and after distal and proximal co-culture (Figure 6e-h, Supplementary Figure 7b-c). We have analyzed the expression of a list of 250 proximal and distal mesenchymal markers, described by Wang et al., 2018³, in our co-cultured iLM. We observe an increased expression of markers described as distal by Wang et al. after distal co-cultures, as well as an enrichment of markers described as proximal by Wang et al after proximal co-culture (Figure 6f, Supplementary Figure 7c). We have added the data showing our new single cell RNAseq dataset in Figure 6 and Supplementary Figure 7 and further discuss our findings line 499-510.

6. One key missing piece in this manuscript is whether iLMs can generate mature lung mesenchymal cells found in adult lungs? Although the authors showed expression of a couple of markers by RT-PCR or immunostaining, some of the markers tested in this manuscript do not

distinguish distinct lung mesenchymal cell populations. The authors compared their scRNA-seq data with that of Zepp et al and claim that they could identify MANCs, ASM, and VSMs. While Zepp et al., have identified some sub-populations of mesenchymal cells, recent studies including Tsukui et al. have provided additional subsets of adult mouse lung mesenchymal cells and associated markers. These recent studies identified per-bronchiolar fibroblasts, adventitial fibroblasts, alveolar (interstitial) fibroblasts etc. Therefore, this reviewer suggest that the authors compare their data with Tsukui et al. Nature communications paper.

We thank the reviewer for this suggestion and have now added these comparisons. We agree that our iLM (both pre and post co-culture) should be compared to other sub-populations of mesenchymal cells of the lung. As mentioned in the previous comment, we have now performed scRNA-seq of our iLM before and after proximal and distal co-culture. We have also added gene lists containing the top50 enriched genes for the populations identified both by Zepp et al.⁴ and Tsukui et al.⁵ to this scRNA-seq dataset. As suggested by our RT-qPCR, for the gene lists by Zepp et al. we find an enrichment of MANC and ASM markers after co-culture, whereas VSM markers are expressed at low levels (Figure 6g), confirming our previous findings that cells seem to differentiate towards MANC- and ASM-like cells, but not towards the VSM. For the populations described by Tsukui et al. we find little to no expression in iLM before co-culture, confirming that iLM prior to recombinant culture, seems to represent an immature progenitor state. In contrast, after recombinant culture we observe increased expression of markers for several of the cell types characterized by Tsukui et al., in particular peribronchial, alveolar and adventitial fibroblasts (Figure 6h), suggesting that co-cultured iLM differentiates towards these more mature cell types. Importantly, we would like to emphasize that while our data suggests that there is some degree of differentiation of our iLM after co-culture, these cells do not necessarily represent fully mature mesenchymal cells, and future work will need to focus on the maturation of the various sub-lineages of lung mesenchyme. We have added additional figures showing this new data and have edited the manuscript to discuss these new findings (line 510-520 and 587-591).

Reviewer #2 (Remarks to the Author):

Alber et al. established a three-step protocol to selectively induce the lung mesenchyme (LM) from mouse iPS cells harboring GFP under the control of a lung-specific Tbx4 enhancer. After analysis of the induced lung mesenchyme (iLM) by scRNA-seq, the authors combined the iLM with ES cell-derived lung epithelia to show the increased yields of epithelia in the organoids, along with increases in some mesenchymal genes. The combination of iLM with ES cell-derived endodermal cells also enhanced the yields of lung epithelial cells.

Although the approach of using LM-specific GFP is useful, the study does not strictly address similarities between the iLM and its in vivo counterpart, or the superiority of the iLM over the conventional lung mesenchyme co-induced with lung epithelia (cLM). Furthermore, the study does not include functional assays involving organoids generated by combining lung epithelia with cLM or LM in vivo. In the current form, it remains unclear how similar the iLM is to the LM

in vivo; it also remains unclear whether the iLM is superior to the cLM.

Major comments

1. It remains unclear how similar the iLM is to its in vivo counterpart or whether the iLM is superior to the cLM. The authors should merge all scRNA-seq data (GFP-positive iLM, GFP-negative iLM, cLM, and E10.5&12.5 LM in vivo) using Seurat software; they should present UMAP plots that show the overlap of the iLM cluster, but not the other clusters, to the in vivo clusters. Theoretically, E12.5 LM should separate into multiple clusters, including immature progenitors, proximal and distal mesenchyme, smooth muscles, and many others. I would like the merged plots to be shown with better resolution. It would also be informative to perform hierarchical clustering analysis and/or calculate Pearson correlation coefficients to show quantitative similarities and differences between the compared samples. Similarly, organoids generated by combining iLM and lung epithelia should also be analyzed by scRNA-seq and merged in the UMAP plots described above.

We thank the reviewer for this suggestion and we have now added a more detailed computational analysis as well as new experiments and new scRNA-seq profiles of our recombinant cultures based on the reviewer's thoughtful input.

First, as suggested by the reviewer we now show a combined SPRING plot of our iLM, iLM^{Neg}, cLM and E12.5 primary populations including a higher-resolution Louvain clustering analysis (res 0.5). As predicted by the reviewer, using this higher resolution clustering our primary cells separate into various lung mesenchymal sub-populations, which have also been identified in E12.5 lung mesenchyme by Zepp et al., 2021⁴, such as mesenchymal progenitors, smooth muscle cells, tracheal mesenchyme and mesothelium (Supplementary Figure 2a).

We agree with the reviewer that it is important to further quantify the similarity of iPSC-derived lung mesenchyme to primary embryonic lung mesenchyme and to co-developed lung mesenchyme. While Pearson correlations can be useful to compare similarities between different cell populations, we find that in our context, due to the mesenchymal nature of all our cell populations all Pearson correlations are very high and do not allow interpretable measures of differences between populations, since mesenchymal genes dominate (>0.96 correlations between all comparisons). Instead of showing Pearson correlation plots we have thus included an alternative method of quantification, where we show the combined expression of the LgM and Han et al. lung mesenchymal marker gene sets in our iLM, iLM^{Neg}, cLM and Primary populations as boxplots (Supplementary Figure 2f). These boxplots illustrate that both gene sets are expressed at significantly higher levels in iLM cells compared to cLM cells and iLM^{Neg} cells. In addition, using unsupervised hierarchical clustering, we have generated a dendrogram of our iPSC-derived and primary populations and find that the iLM population clusters closer to our primary lung mesenchyme than iLM^{Neg} or cLM cells (Figure 2d), suggesting that iLM cells are more similar to primary lung mesenchyme than cLM cells. We have adjusted the manuscript to discuss these various points.

We have also added additional time series RT-qPCR analyses of a variety of time points from early lung mesenchymal development in vivo (E11.5-18.5), providing comparisons to our iLM, added as new Supplementary Figure 3e. Importantly, we would like to underline here that we use E12.5 lung mesenchyme as our earliest primary timepoint for scRNA-seq analysis, because we found it to be experimentally unfeasible to collect enough E10.5 lung mesenchyme to perform meaningful analysis. The number of cells activating the Tbx4-LER reporter in vivo at earlier time-points is very low, leading to low yield of cells upon sorting. Furthermore, we would like to clarify that we do not expect any of our engineered clusters to overlap with primary cells. Our lab and others have previously shown that, despite similar expression of key marker genes, iPSC-derived cell populations and their in vivo counterparts tend to not overlap on SPRING/UMAP plots (see for example Alysandratos et al., JCI 2022⁶), unless visualized with techniques such as “harmonization” (designed to overcome technical batch effects), which tends to make the engineered and primary populations overlap, and can make populations appear more similar than they actually are. Thus, in this manuscript, we have worked hard to avoid harmonization artifacts or over-correction by analyzing all samples in parallel thus avoiding technical batch effects, and preserving their expected clustering differences. Some of the clustering differences might be explained in part by differences unrelated to their lung-mesenchyme specific identity, such as differences in their proliferative state or differences in gene expression driven by their substrate/surrounding niche.

Furthermore, as suggested by the reviewer we have performed an additional scRNA-seq experiment analyzing our organoid/recombinant co-cultures of iLM and epithelial progenitors in distal and proximal culture conditions (Figure 6). In order to compare this new scRNAseq dataset with primary lung mesenchyme from various time-points we show a merged UMAP plot containing our iLM cells before and after co-culture and a dataset by Zepp et al., 2021⁴, consisting of embryonic, postnatal and adult lung mesenchyme (Supplementary Figure 8). Our new scRNA-seq dataset suggests further mesenchymal differentiation after both proximal and distal co-culture, even though our cells do not represent fully mature/adult cells after 1 week in co-culture. Importantly, we would also like to underline that our merged UMAP plot should be interpreted with caution, since the two datasets were generated by different labs and on different machines, leading to potential batch effects that might contribute to differential cluster and gene expression profiles.

2. However, I suspect that the iLM may not be similar to its in vivo counterpart, because Fig. 2 shows significantly lower expression levels of Tbx4 and Wnt2, while Wnt2b is de-repressed. To achieve significant overlaps in the UMAP plots, the expression levels of most key genes would need to reach the levels observed in vivo. The increased expression levels of Shh downstream genes are presumably caused by Shh treatment, which does not serve as evidence of similarity. Because Fig. 1e shows the essential role of Wnt for GFP expression, additional Wnt activation by Wnt3a or CHIR99021 may be helpful. It may also be helpful to optimize the concentrations of all reagents. It is important to compare the expression levels of multiple genes in the iLM to the levels in the in vivo counterpart at multiple time points. GFP is a useful monitoring tool, but strict reliance on GFP percentages may provide misleading evidence concerning induction.

We agree and have revised our manuscript in response to these concerns. As correctly predicted by the reviewer, our data suggest differential clustering and differences in expression of certain lung mesenchymal markers comparing primary E12.5 primary lung mesenchyme to iLM, with *Wnt2* expression being an important difference between these populations (Figure 2h, Supplementary Figure 2d; as we have emphasized in our original submission), whereas other lung mesenchymal markers from the LgM and Han et al. gene sets exhibit overlapping expression. As suggested by the reviewer, we include new experiments with scRNA-seq profiles of recombinant organoids after co-culture of our iLM with epithelium, indicating significantly upregulated *Wnt2* in distal conditions (Figure 7e). As discussed in the previous point, we do not expect our iLM to completely overlap with primary lung mesenchyme, but we do think that the expression of many key markers of embryonic lung mesenchyme suggests that our iLM is lung mesenchyme-like, despite some markers being expressed at lower levels. We do not want to give the impression that we think our engineered mesenchyme is equivalent to primary mesenchyme, and we have attempted to make these points more clear in our revised manuscript in response to the reviewer's questions, particularly emphasizing differences between engineered LM and primary control tissue. We have also added new experimental data in order to address the reviewer's comments. We respond stepwise to each helpful suggestion or question posed above, as follows:

Because Fig. 1e shows the essential role of Wnt for GFP expression, additional Wnt activation by Wnt3a or CHIR99021 may be helpful.:

First, we have more systematically assessed the role or necessity of canonical Wnt stimulation/activity by adding either the Wnt inhibitor XAV or two different concentrations of recombinant WNT3A to our minimal culture medium (Figure 1f). We find that adding XAV significantly decreases the percentage of GFP+ cells, suggesting that successful lung mesenchyme differentiation requires active Wnt signaling (endogenous). In contrast, stimulating Wnt signaling with two varying concentrations of WNT3A (exogenous supplementation) did not significantly alter the percentage of GFP+ cells. We have updated our text results to reflect these new results and their interpretations (line 163-167).

The increased expression levels of Shh downstream genes are presumably caused by Shh treatment, which does not serve as evidence of similarity :

We agree that the upregulation of Shh targets is expected upon treatment with the Hh agonist present in the medium. To expand our panel of markers that are not direct targets of the Hh or RA pathways, we have now added 9 additional early lung mesenchymal markers found by Han et al., 2020 to our RT-qPCR analysis (Figure 2i, Supplementary Figure 3c). We do not detect any significant differences in the expression of any of these markers between iLM and primary lung mesenchyme, whereas 3/9 markers are expressed at higher levels iLM compared to iLM^{Neg} cells, further illustrating that despite certain differences in gene expression, many lung mesenchymal markers are expressed at similar levels in iLM and primary lung mesenchymal cells.

It may also be helpful to optimize the concentrations of all reagents.:

We agree and have now included additional figures showing concentration curves for RA and PMA and the dose response of our reporter in an effort to better explain the concentrations we use in our protocol, concentrations that have been optimized for optimal yield of Tbx4-LER^{GFP+}

cells (Supplementary Figure 1i, j). We show that the percentage of GFP+ cells reaches a plateau after 1uM of RA, while the yield (i.e. number) of GFP+ cells peaks at 2uM. Furthermore, the percentage of GFP+ cells peaks at 1ug/ml of PMA. However, we find that PMA concentrations higher than 0.5ug/ml lead to detachment and death of cultured cells, reducing the yield of GFP+ cells. We thus conclude that using 0.5ug/ml of PMA yields maximal numbers of GFP+ cells. We have updated our results text accordingly (line 170-171).

It is important to compare the expression levels of multiple genes in the iLM to the levels in the in vivo counterpart at multiple time points:

As suggested we have added a RT-qPCR time-course analysis for several lung mesenchymal markers known from the literature and from the gene list by Han et al., 2020 as well as more mature and general mesenchymal markers, sorting Tbx4-LER^{GFP}+ cells every day from day 8 – 13 of differentiation and also isolating primary lung mesenchymal cells from embryonic day 11.5 (i.e. the earliest time-point at which it is technically feasible to collect enough cells for RT-qPCR analysis), 12.5, 13.5 and 18.5 (Supplementary Figure 3e). Importantly, in contrast to the RT-qPCR data shown in Figure 2h-i, for our directed differentiation we only induced our Tbx4-LER reporter 2 days before collecting the cells in order to only isolate cells that have an active reporter in the time immediately before harvesting. We find different temporal dynamics for different lung mesenchymal markers. For the directed differentiation, a subset of markers, such as *Tbx4*, *Hmga2*, and *Rspo1*, decrease over time, whereas other markers, such as *Ptch1*, *Pde5a* or *Tnfrsf19* remain expressed at similar levels over all time-points. Finally, the expression of many markers seems to peak around day 10-12 and slightly decrease on day 13. We conclude that collecting cells on day 11-12 of differentiation is optimal, due to the peak in expression levels of many lung mesenchymal markers, as well as due to the higher yield that can be achieved on those days. Based on these findings, all co-culture experiments (Figure 3-6) have been performed using iLM from day 11 of differentiation (with addition of doxycycline on day 9). Furthermore, while we do observe certain fluctuations in the expression levels of lung mesenchymal genes between different embryonic time-points, expression levels of most markers seem to be similar in cells from in vivo embryonic day E11.5 and 12.5. Given the limited availability of cells from earlier time-points we thus conclude that embryonic day 12.5 cells are an adequate control for scRNA-seq, RT-qPCR, and co-culture experiments.

3. Another possible reason for the insufficient gene expression is that the proposed protocol does not accurately consider the corresponding embryonic stages in vivo. Under the assumption that ES cells represent the E3.5 blastocyst stage, I wonder whether it is reasonable to conclude that ES cells differentiated for 5 days and 13 days correspond to E8.5 and E12.5, respectively. In particular, STEP 3 lasted for 8 days with a fixed condition. Perhaps better quality can be achieved by shortening STEP 3 or subdividing it into multiple steps. Overall, the authors should clarify which stage they are targeting (e.g., E10.5, E12.5, or later).

We thank the reviewer for this comment and have edited the text accordingly. We would like to clarify that the ideal stage that we would like to target for the purposes of this paper (initial induction of developing lung specific mesenchymal progenitors) is soon after early lung mesenchyme specification (from embryonic up to pseudoglandular stages), i.e. E9-E12 (as

defined in Rackley and Stripp, JCI 2012⁷). In response to the reviewer's concerns we have now added new experiments and data from a time series analysis comparing gene expression in our engineered iLM to that of multiple in vivo primary developing lung mesenchymal time points (E11.5, E12.5, E13.5 and E18.5, Supplementary Figure 3e). Importantly, it is not technically feasible for us to obtain enough single cell lung mesenchyme for RT-qPCR earlier than day E11.5. As discussed in comment #2 our new in vitro time-course experiment provides parallel RT-qPCR analyses on every day of differentiation from in vitro day 8 on. We find that the expression of many lung mesenchymal markers does not change substantially over time, whereas the expression of others peaks around day 10-12. We thus used day 11 cells for all co-culture experiments.

Regarding the question of the duration of each in vitro stage: we would like to emphasize that the *sequence* of milestones or developmental fate decisions in ESC/iPSC in vitro directed differentiations is often similar to in vivo sequences, but the precise *duration* of each stage substantially differs from in vivo development timing. For example, human alveolar epithelial type 2-like cells can be derived from human iPSCs in 4-5 weeks, but in vivo development of the equivalent milestone in vivo takes 24 weeks (Jacob et al., 2017⁸).

4. Notably, Wnt2 is upregulated in the iLM, upon combination with ES cell-derived lung epithelia; this finding suggests crosstalk between the mesenchyme and epithelia. This is also related to Comment 3. If the iLM corresponds to the stage when Wnt2 remains low (e.g., E10.5), the crosstalk may upregulate Wnt2 expression to the level observed at E12.5. Thus, it is essential to compare the expression levels of key genes in the iLM and cultured organoids to those expression levels within in vivo samples at various stages (e.g., E10.5 and 12.5), using qPCR and eventually scRNA-seq.

We agree and have added new experiments and analyses in response to these suggestions, finding that indeed *Wnt2* significantly increases after co-culture with epithelium. As discussed in previous comments, we have added a time-course experiment, analyzing iLM from various time-points and primary lung mesenchyme from E11.5-E18.5. In addition, we have performed a scRNA-seq experiment (Figure 6, Supplementary Figure 7) analyzing co-cultured iLM recombinants. Importantly, our scRNAseq data confirms the up-regulation of *Wnt2* expression in co-cultured iLM (Figure 6e).

5. Although the authors used MEFs as negative controls for functional assays (organoid generation), they should use the cLM to demonstrate the advantages of their new induction protocol in the main experiments in Figs. 4–7. Together with scRNA-seq analysis (Comment 1), the findings would indicate whether the quality or yield of the induced cells have improved. It is also important to include the LM in vivo for organoid generation as an ideal reference to clarify the functional similarity of iLM to its in vivo counterpart. For example, Fig. 4 shows the disturbed differentiation of epithelial cells when combined with iLM, but it is unclear whether this occurs when combined with the LM in vivo. The authors should also explore the effects of exogenous cytokine omission on the organoids. Ref. 31 showed the dominant role of the LM in the fate determination and differentiation of lung epithelia. If this dominant role is a consistent

phenomenon, the genuine mesenchyme should direct proximal or distal lung fate and promote epithelial differentiation by providing key signaling factors. However, the authors added exogenous cytokine cocktails to control such fates and the iLM appeared to antagonize such differentiation, rather than eliminating the need for mesenchyme-derived factors. The authors should demonstrate that the observed phenotypes represent physiological processes.

Please see our response to point #1 above where we have included more quantitative cLM vs iLM comparisons by scRNA-seq in response to this concern. In any experimental approach, one needs to consider appropriate controls and we certainly appreciate the pros and cons, such as those raised by the reviewer of selecting MEFs vs cLM vs other types of mesenchyme as controls for our experiments. We have made an effort to respond to the reviewer's suggestion with attempts to generate sufficient numbers of cLM cells by co-development and find it difficult to generate sufficient numbers of cLM cells for recombinant cultures (indeed this is one reason why engineering a dedicated, expandable source of lung mesenchyme, e.g. iLM, is useful for developmental studies.) The end result is it is not technically possible for us to repeat the vast number of experiments across 4 main figures with cLM as a control cell type, given the low yield of cLM. For this reason, we have used an alternative type of primary developing mesenchyme, MEFs, as a control for those figures, but we have also added new experiments with primary lung mesenchyme controls in later figures (detailed below), as suggested by the reviewer. We hope the reviewer can appreciate our reasons for selecting MEFs as a control, particularly given our shared priority of trying to work with primary cells as a control.

We do agree that co-culture with primary lung mesenchyme as a positive control would be very interesting and have added this experiment in the revised version of the manuscript (Figure 5a-c, Supplementary Figure 5d). We find that E12.5 lung mesenchyme can be co-cultured with engineered epithelial progenitors in distal conditions and cells self-organize in a similar manner. We show by immunofluorescence that primary lung mesenchymal cells can be found in close juxtaposition to engineered epithelial progenitors (Figure 5b). We have also added data where we co-culture engineered lung epithelial progenitors with either iLM or primary embryonic lung mesenchyme in parallel in distal medium and obtain similar yields of Nkx2-1^{mCherry+} cells after one week of co-culture in distal conditions (Figure 5c). Finally, when quantifying gene expression in these co-cultures we find no significant differences in expression of *Etv5*, *Sox9* and *Sftpc* in resorted epithelial cells (Supplementary Figure 5d). In contrast, we do observe higher levels of *Ager* transcripts.

Finally, we have performed an experiment where we co-culture iLM with ESC-derived lung epithelial progenitors removing either one media factor at a time, or all growth factors and have included this data in the manuscript (Supplementary Figure 5e). Of note, we do not observe any significant outgrowth when culturing lung epithelial progenitors alone, with iLM or with primary LM in serum-free differentiation medium without any growth factors, suggesting that cultured cells require at least a minimal presence of growth factors in the medium. This is in agreement with the current literature, since we do not know of any successful (co-)culture of lung epithelial cells without any growth factors or serum-containing medium. We respectfully point out that the classic recombinants of Shannon referred to in Shannon, 1994⁹ and Hawkins et al., 2017¹⁰ were

not truly cytokine/growth factor free as they were performed with the addition of factors: both serum and recombinant Hh. Thus, instead of co-culturing cells without any growth factors at all (which does not allow cell survival) we removed one factor from the medium at a time in order to investigate whether our iLM can compensate for the loss of certain growth factors. We quantified the yield of Nkx2-1^{mCherry+} cells upon growth factor removal and find our iLM can partially compensate for the removal of WNT3A and FGF2 from the distal medium and for the removal of FGF10 and FGF2 from the proximal medium (Supplementary Figure 5e). We have added these new experiments to our revision.

6. The authors sometimes compared iLM with adult LM, which is not valid. For example, Ref. 41 showed that AT2 cells can be maintained in the presence of adult LM without exogenous factors. However, the authors combined AT2 cells with iLM, then added various cytokines to show the survival of AT2 cells (Fig. 6). This approach does not prove that the functionality of iLM is similar to that of adult LM. Fig. 7 also shows analysis of the adult lung; the obtained markers were used for qPCR analysis of the iLM-containing organoids. To support the authors' claims, scRNA-seq analysis should be utilized to compare the differentiation/maturation states of the iLM-containing organoids with the adult lung or—more realistically—embryonic lung.

We agree with these comments and have revised our manuscript and added new experimental data in response. As suggested by the author, we have performed a scRNA-seq experiment using our iLM-containing organoids and have added further comparisons to mature lung mesenchymal cell types in order to investigate the maturation state of our iLM before and after co-culture. We would like to underline that we do not believe that our iLM is adult-like, and we have edited our interpretations to more clearly reflect this. Both our RT-qPCR and scRNAseq analysis suggest that iLM resembles an early lung mesenchymal progenitor cell. We show that, upon co-culture in distal or proximal medium the expression of several sets of markers for mature lung mesenchymal cell types increases. However, we do not think that these cells are fully mature at this point, and again we have revised our manuscript to make this clear (lines 587-591).

We use co-cultures of iLM with adult primary AT2 cells simply as an additional approach to demonstrate that iLM can function by supporting AT2 cell survival and outgrowth in culture, a cell type that cannot be maintained ex vivo by itself in most published media to date, including our medium. However, we completely agree that these experiments do not suggest that iLM is equivalent or similar to mature lung mesenchyme and have edited our manuscript to more clearly emphasize this, in response to the reviewer's concerns. Our iLM is just one of a long list of mesodermal/mesenchymal cells of varying developmental stages able to support primary AT2 cell cultures, including Mlg fibroblasts, MRC5 fetal fibroblasts, PDGFRa adult fibroblasts, and monocytes.

Conversely, we also tested primary fetal lung epithelial progenitors in co-cultures with our iLM (see also minor comment #2) and have used RT-qPCR in order to investigate whether co-culture with iLM similarly increases expression of immature epithelial markers and decreases the

expression of mature epithelial markers in co-cultured primary epithelial cells from embryonic day 12.5 (Figure 5g-j). We find that, similar to our co-cultures with engineered lung epithelium, co-culture of primary fetal lung epithelial Nkx2-1^{GFP}+ cells with iLM in distal medium significantly increases the yield of Nkx2-1^{GFP}+ progenitors compared to Nkx2-1^{GFP}+ cells cultured in distal medium alone. In addition, we find increased expression of the early lung epithelial marker *Etv5* and decreased expression of the more mature epithelial lineage marker *Sftpc* in primary embryonic epithelial cell that were co-cultured with iLM, in agreement with the observations in our co-culture with engineered epithelium.

Minor comments

1. Fig. 3 simply shows that the iLM has generic mesenchymal stem cell-like properties; it does not strengthen the authors' claim that the iLM is similar to the LM *in vivo*.

We completely agree and have edited our manuscript to indicate that this figure simply demonstrates the mesenchymal phenotype of our engineered iLM, not any lung specific qualities as it was not our intention to use this figure for anything more than evidence of general mesenchymal competence. Out of respect for the reviewer's point, we have also moved this figure (old main Figure 3) to the supplement (now Supplementary Figure 4) as it represents a minor part of our story and is not meant to support "lung specificity".

2. The authors used ES cell-derived lung epithelial cells for organoid generation. However, because the quality of ES cell-derived cells is not completely equal to the quality of their *in vivo* counterparts, primary epithelial cells from transgenic mice may sometimes be more useful for rigorously testing the functional competence of the induced LM.

We agree and thank the reviewer for the suggestion. As described above in comment #6 we have now added an experiment where we co-culture our iLM with primary lung epithelial progenitors from E12.5 in distal medium (Figure 5g-j).

References

1. Kwong, G., Marquez, H. A., Yang, C., Wong, J. Y. & Kotton, D. N. Generation of a Purified iPSC-Derived Smooth Muscle-like Population for Cell Sheet Engineering. *Stem Cell Reports* **13**, 499–514 (2019).
2. Han, L. *et al.* Single cell transcriptomics identifies a signaling network coordinating endoderm and mesoderm diversification during foregut organogenesis. *Nat Commun* **11**, 4158 (2020).
3. Wang, C. *et al.* Expansion of hedgehog disrupts mesenchymal identity and induces emphysema phenotype. *Journal of Clinical Investigation* **128**, 4343–4358 (2018).
4. Zepp, J. A. *et al.* Genomic, epigenomic, and biophysical cues controlling the emergence of the lung alveolus. *Science* **371**, eabc3172 (2021).
5. Tsukui, T. *et al.* Collagen-producing lung cell atlas identifies multiple subsets with distinct localization and relevance to fibrosis. *Nat Commun* **11**, 1920 (2020).
6. Alysandratos, K.-D. *et al.* Culture impact on the transcriptomic programs of primary and iPSC-derived human alveolar type 2 cells. *JCI Insight* **8**, e158937 (2023).

7. Rackley, C. R. & Stripp, B. R. Building and maintaining the epithelium of the lung. *J. Clin. Invest.* **122**, 2724–2730 (2012).
8. Jacob, A. *et al.* Differentiation of Human Pluripotent Stem Cells into Functional Lung Alveolar Epithelial Cells. *Cell Stem Cell* **21**, 472-488.e10 (2017).
9. Shannon, J. M. Induction of Alveolar Type II Cell Differentiation in Fetal Tracheal Epithelium by Grafted Distal Lung Mesenchyme. *Developmental Biology* **166**, 600–614 (1994).
10. Hawkins, F. *et al.* Prospective isolation of NKX2-1–expressing human lung progenitors derived from pluripotent stem cells. *Journal of Clinical Investigation* **127**, 2277–2294 (2017).

REVIEWERS' COMMENTS

Reviewer #1 (Remarks to the Author):

I have no further comments

Reviewer #2 (Remarks to the Author):

I expected the authors to modify the protocol to improve the quality of the iLM, but instead they preferred to support the validity of the current induction conditions. I appreciate that they have addressed many of my concerns. My remaining concern is the inadequate quality of the iLM compared to the in vivo LM: the expression levels of key genes such as *Tbx4*, *Wnt2*, *Wnt2b*, and *Nkx6.1* are very different from those in vivo. The authors acknowledge this in their response letter, but it is not fully reflected in the manuscript. For example, for Fig. 2h and the newly added time course experiment (Supplementary Fig. 3e), the authors do not mention this point in the Results section. In Fig. 2b-e, they only briefly mention that the SPRING/UMAP clusters do not overlap without explaining that this means that the iLMs are quite different from the in vivo LMs. Cells with similar gene expression should be classified in the same cluster even in a standard Seurat analysis without "harmonization". On page 10, the paragraph title "Induced lung mesenchyme is transcriptionally similar to primary embryonic lung mesenchyme" is retained. In the Discussion section (page 28, second paragraph), it is unclear how the increase in *Wnt2* levels by co-culture is equivalent to in vivo. At the very least, the authors should describe the weaknesses of the data more clearly and tone down the claims in the manuscript to the extent of the response letter.

We thank reviewer 2 for the additional input and have further revised our manuscript accordingly. Our responses point-by-point are as follows, highlighted in blue font following the reviewer's comment below and also highlighted in yellow in the revised manuscript:

Reviewer 2 comments:

"I expected the authors to modify the protocol to improve the quality of the iLM, but instead they preferred to support the validity of the current induction conditions. I appreciate that they have addressed many of my concerns. My remaining concern is the inadequate quality of the iLM compared to the in vivo LM: the expression levels of key genes such as *Tbx4*, *Wnt2*, *Wnt2b*, and *Nkx6.1* are very different from those in vivo. The authors acknowledge this in their response letter, but it is not fully reflected in the manuscript. For example, for Fig. 2h and the newly added time course experiment (Supplementary Fig. 3e), the authors do not mention this point in the Results section. In Fig. 2b-e, they only briefly mention that the SPRING/UMAP clusters do not overlap without explaining that this means that the iLMs are quite different from the in vivo LMs. Cells with similar gene expression should be classified in the same cluster even in a standard Seurat analysis without "harmonization". On page 10, the paragraph title "Induced lung mesenchyme is transcriptionally similar to primary embryonic lung mesenchyme" is retained. In the Discussion section (page 28, second paragraph), it is unclear how the increase in *Wnt2* levels by co-culture is equivalent to in vivo. At the very least, the authors should describe the weaknesses of the data more clearly and tone down the claims in the manuscript to the extent of the response letter."

We thank the reviewer for these comments and have modified the manuscript text to further highlight the differences in expression of multiple lung mesenchymal genes between iLM and primary lung mesenchyme:

First, we have modified the results text to point out genes that were expressed at different levels in iLM and primary controls. Our manuscript now contains text further emphasizing the following differences between iLM and primary mesenchyme:

On page 11, line 228 we are careful to emphasize *Wnt* signaling differences between iLM and primary control mesenchyme, as follows: "*Next, focusing on differences between iPSC-derived cluster 1 cells and primary cells, we found Wnt2 to be more highly expressed in primary cells, along with canonical Wnt signaling target genes, such as Lef1 and Snai1/2 (Supplementary Fig. 2b,d), suggesting that activation of the Wnt signaling pathway is lower in cluster 1 cells in these conditions compared to primary embryonic lung mesenchyme.*"

On page 14, line 285 we have added text to emphasize differences, as follows: "*Importantly, while many lung mesenchymal markers were expressed at similar levels in iLM and primary lung mesenchyme, we did detect significant differences in expression of a subset of markers. For example Tbx4 and Wnt2 were expressed at significantly lower levels in iLM cells compared to E12.5 lung mesenchyme, whereas the expression of Wnt2b was significantly higher. Consistent with these results the total module scores of both Han et al and LgM gene sets were slightly lower in iLM than in primary E12.5 mesenchyme (Supplementary Fig. 2f).*"

On page 15, line 311 we discuss differences in gene expression in our new time-course experiment as follows: "*Expression levels of 12 out of 15 markers were similar to E11.5-12.5 primary lung mesenchyme, but we observed important differences in Wnt2 and Tbx4, which were lower in the Tbx4-LER^{GFP+} cells at all time points assayed (Supplementary Fig. 3e), consistent with our scRNA-seq dataset (Supplementary Fig. 2 b-d).*"

Second, we have edited the discussion to acknowledge that there are important transcriptomic differences between iLM and primary cells (page 25, line 547-559). As mentioned in our previous response letter, we do not expect primary and iPSC-derived cells to overlap in SPRING/UMAP

plots. We do agree that while part of these differences might be caused by cell culture related effects, at least part of the separate clustering might be caused by differences in the expression of lung mesenchymal marker genes, and we have added a corresponding paragraph in the discussion to emphasize this.

Third, as suggested by the reviewer, we have modified the paragraph title “Induced lung mesenchyme is transcriptionally similar to primary embryonic lung mesenchyme” to “Induced lung mesenchyme expresses multiple lung mesenchymal markers”.

Finally, regarding the levels of expression of *Wnt2* after co-culture: While the RT-qPCR experiments in Figure 2h (i.e. comparing *Wnt2* expression in iPSC-derived and primary cells) and 6a (i.e. showing *Wnt2* expression after co-culture) were not performed on the same plate, we did use the same reagents, machine and day 0 cDNA for normalization. Thus, we do believe that the rough range of expression is comparable and seems similar in primary cells and co-cultured lung mesenchyme (i.e. 2000-3000 fold change over day 0 iPSCs).

In conclusion, we have adjusted the text to be more reflective of the response letter according to the reviewer’s comments, focusing in particular on both the similarities and important differences between iLM and primary lung mesenchyme.